# FGFR Aberrations in Solid Tumors: Mechanistic Insights and Clinical Translation of Targeted Therapies

**DOI:** 10.3390/cancers18010089

**Published:** 2025-12-27

**Authors:** Zijie He, Yizhen Chen, Genglin Li, Jintao Wang, Yuxin Wang, Pengjie Tu, Yangyun Huang, Lilan Zhao, Xiaojie Pan, Hengrui Liu, Wenshu Chen

**Affiliations:** 1Department of Thoracic Surgery, Shengli Clinical Medical College of Fujian Medical University, Fujian Provincial Hospital, Fuzhou University Affiliated Provincial Hospital, 134 East Street, Fuzhou 350001, China; hezijie0824@fjmu.edu.cn (Z.H.); cyz765488791@126.com (Y.C.); lgl18065003992@fjmu.edu.cn (G.L.); wangjt690@fjmu.edu.cn (J.W.); 3211003371@stu.fjmu.edu.cn (Y.W.); tupengjie@tmu.com (P.T.); thoracichyy@163.com (Y.H.); zhaolilan@fzu.edu.cn (L.Z.); xyr8837924@126.com (X.P.); 2Department of Oncology, University of Cambridge, Cambridge CB2 1SZ, UK

**Keywords:** FGFR aberrations, targeted therapy, receptor tyrosine kinases, oncogenic signaling, therapeutic resistance

## Abstract

Cancer treatment is increasingly guided by tumor DNA alterations. Changes in fibroblast growth factor receptors, a family of four related proteins, occur across many solid tumors and drive growth, spread, and treatment response by activating multiple signaling pathways. This review examines how these receptor changes—amplifications, mutations, and fusions—differ in their mechanisms and distribution across cancer types, and how they reshape the tumor environment, blood supply, and immune landscape. We summarize the clinical benefits and limitations of approved targeted drugs, explain why resistance develops through secondary mutations or alternative signaling routes, and describe how blood-based tests may detect resistance early. We also discuss next-generation strategies including combination with immunotherapy or anti-vessel therapy, antibody–drug conjugates, and nanotechnology-based delivery systems. By integrating molecular, translational, and clinical evidence, this review aims to guide patient selection, treatment monitoring, and the design of durable precision therapies targeting these receptor abnormalities.

## 1. Introduction

Malignant tumors represent the second leading cause of death worldwide. GLOBOCAN 2022 reported 20.1 million new cancer cases and 9.7 million cancer-related deaths globally in 2022. Projections indicate that new cancer cases will reach 28.4 million by 2040 [1]. Pan-cancer multi-omics studies have systematically identified widespread aberrations in key oncogenic drivers, including *TP53*, *KRAS*, and *PIK3CA* [2]. These discoveries have accelerated the development of precision oncology.

Cancer treatment has shifted from traditional surgery, radiotherapy, and chemotherapy toward comprehensive strategies guided by molecular biomarkers. The fibroblast growth factor receptor (FGFR) family comprises four receptor tyrosine kinases (FGFR1–4) that regulate cell proliferation, differentiation, and angiogenesis. Upon ligand binding, these receptors activate the RAS/MAPK, PI3K/AKT, and PLCγ signaling pathways. This activation drives tumor growth and remodels the tumor microenvironment [3]. Regulatory agencies have approved pemigatinib for cholangiocarcinoma and erdafitinib for urothelial carcinoma. Additionally, futibatinib has demonstrated promising selectivity and tolerability in clinical trials [4,5,6].

FGFR aberrations exhibit distinct distributions across cancer types. These aberrations include amplifications, mutations, fusions, and aberrant splicing. *FGFR2* fusions occur in approximately 10–16% of intrahepatic cholangiocarcinomas [7,8]. In contrast, *FGFR3* mutations are present in 17–20% of bladder cancers [9,10]. These genetic alterations drive oncogenic phenotypes. They also promote angiogenesis and facilitate immune evasion. Recent single-cell and spatial omics studies have revealed the critical role of FGFR signaling in tumor heterogeneity and clonal evolution [11].

Despite therapeutic advances, FGFR-targeted therapies face significant clinical challenges. Tumors develop acquired resistance through secondary mutations or bypass signaling activation. Patients experience dose-limiting toxicities, including hyperphosphatemia and cutaneous or ocular adverse events. Furthermore, clinicians face uncertainties in detecting FGFR aberrations across diverse cancer types and determining the druggability of rare variants [12,13,14,15,16]. Next-generation inhibitors, such as tinengotinib and rogaratinib, demonstrate improved selectivity and tolerability in preclinical and early clinical studies [17,18]. These agents may enhance clinical outcomes through optimized drug design and biomarker-driven patient selection.

This review systematically examines FGFR aberrations and oncogenic mechanisms in solid tumors, clinical development, and combination strategies, emphasizing resistance biology and biomarker advances. We propose multi-omics-based predictive frameworks to guide precision application and drug development of FGFR-targeted therapies.

## 2. Molecular Biology and Tumor Distribution of the FGFR Family

### 2.1. Structural and Functional Features of the FGFR Family

The FGFR family belongs to the receptor tyrosine kinase (RTK) superfamily and plays central roles in cell proliferation, differentiation, and metabolism. It comprises four highly conserved isoforms (FGFR1–4) that initiate complex signaling networks upon binding to specific fibroblast growth factors (FGFs), thereby mediating diverse physiological and pathological processes.

#### 2.1.1. Domain Architecture and Molecular Characteristics of FGFR1–4

FGFR1–4 share a highly conserved three-domain architecture: an extracellular domain, a transmembrane region, and an intracellular kinase domain [19]. The extracellular region contains three immunoglobulin-like domains (D1, D2, and D3). D1 and the acid box form an autoinhibitory module, while D2 and D3 constitute the primary ligand-binding region [20]. This structure maintains receptor quiescence in the resting state, with activation occurring only upon ligand binding and dimerization.

In addition to the classical kinase receptors FGFR1–4, the FGFR family includes an atypical member, fibroblast growth factor receptor-like 1 (FGFRL1, also known as FGFR5) [21]. The extracellular domain of FGFRL1 contains three immunoglobulin-like domains and a conserved FGF-binding site, while its intracellular region is limited to a short cytoplasmic tail, lacking the catalytic tyrosine kinase domain [22]. This structural feature classifies FGFRL1 as a “decoy receptor” or “pseudo-receptor.” FGFRL1 binds multiple FGF ligands but does not initiate the classical kinase cascade, thereby modulating local FGF availability, influencing the signaling strength of other FGFRs, and playing a role in tissue morphogenesis and homeostasis [21].

Crystallographic studies reveal that FGFs bind primarily to the D2–D3 region, with heparan sulfate proteoglycans (HSPGs) enhancing binding affinity and promoting dimerization, thereby establishing the molecular basis for FGFR activation [23,24]. The intracellular kinase domain contains an ATP-binding site and multiple tyrosine residues; upon activation, autophosphorylation recruits downstream signaling proteins. Subtle differences in the kinase domain confer distinct substrate specificity and signaling intensity among isoforms [3].

#### 2.1.2. Ligand-Binding Specificity and Signal Transduction Mechanisms

The FGF family comprises 22 members, classified into paracrine and endocrine subfamilies [25]. Paracrine FGFs, such as FGF1 and FGF2, require HSPGs as co-receptors, whereas endocrine FGFs, like FGF19, FGF21, and FGF23, interact specifically with Klotho proteins to modulate FGFR selectivity and activity [26,27]. Klotho proteins act as essential co-receptors that alter FGFR ligand specificity, particularly in tissues such as the liver and kidney. For instance, the FGF19-FGFR4-βKlotho complex enhances binding specificity by stabilizing receptor conformations that preferentially activate FGFR4 signaling in hepatocytes [24,28,29].

Structural studies have revealed that the presence of Klotho proteins influences the affinity of FGFs for FGFR isoforms. For example, FGF19 binding to FGFR4 is greatly enhanced by βKlotho, leading to a higher binding affinity compared to when βKlotho is absent. This alteration in ligand–receptor binding is essential for the regulation of key metabolic pathways such as bile acid synthesis and hepatic glucose metabolism [24,30,31]. On the other hand, Klotho proteins are absent in the binding of FGF1 and FGF2 to FGFRs, which exhibit a broader specificity across multiple FGFR isoforms [32].

Upon activation, FGFR signals predominantly through RAS/MAPK, PI3K/AKT, and PLCγ pathways [33]. FGFR substrate 2 (FRS2) serves as a key adaptor protein, recruiting GRB2 and SOS to initiate the RAS/MAPK pathway for proliferation and differentiation; the PI3K/AKT pathway promotes cell survival and metabolic reprogramming; PLCγ activation modulates intracellular calcium levels and protein kinase C activity [34].

Furthermore, ligand binding to the extracellular Ig-like domains of FGFR leads to receptor dimerization, which is further stabilized in the presence of Klotho proteins. This process is crucial for activating downstream signaling pathways, including RAS/MAPK and PI3K/AKT, which are involved in cell proliferation and survival. The altered specificity induced by Klotho proteins provides a mechanism for the tissue-specific regulation of FGFR signaling, which is pivotal for understanding both normal physiology and oncogenic transformations driven by FGFR aberrations [35,36].

#### 2.1.3. Physiological Functions of the FGFR Family

FGFR signaling is indispensable in both embryonic development and adult homeostasis. During embryogenesis, FGFR1–3 participate in neural, skeletal, and organ formation: FGFR1 is critical for central nervous system and skeletal development, FGFR2 mediates epithelial–mesenchymal interactions, and FGFR3 regulates cartilage and bone growth [37,38].

Beyond the classical FGFRs, genetic and experimental evidence highlights a key role for FGFRL1 in organogenesis, particularly in embryonic kidney and diaphragm development. *Fgfrl1* knockout mice exhibit severe metanephric hypoplasia, often progressing to near-complete renal agenesis, accompanied by a thin, hypoplastic diaphragm that allows for liver herniation, resulting in respiratory failure and perinatal lethality [39,40]. In humans, germline loss-of-function or deleterious *FGFRL1* variants—arising from 4p16.3 terminal deletions or biallelic missense mutations—have been linked to congenital diaphragmatic hernia, though consistent extra-diaphragmatic features, such as cardiac or skeletal anomalies, remain sparsely documented [41]. Mechanistically, FGFRL1 interacts with FGF8 to modulate FGF–FGFR signaling during nephrogenesis, explaining the renal agenesis in null mice, while similar pathways in diaphragmatic myogenesis await elucidation. High FGFRL1 expression in musculoskeletal and renal tissues from embryonic to postnatal stages suggests potential contributions to adult tissue homeostasis and repair, a notion meriting further exploration via single-cell and spatial transcriptomics [39,40,41].

In adult tissues, FGFR pathways maintain homeostasis and participate in repair processes. For example, FGF2–FGFR1 promotes endothelial cell proliferation and migration to regulate angiogenesis; FGF7 stimulates epithelial cell proliferation and differentiation via FGFR2-IIIb, contributing to skin and lung repair [21,33]. Additionally, endocrine FGFs achieve tissue-specific functions in cooperation with Klotho family co-receptors: the FGF19–β-Klotho–FGFR4 axis regulates bile acid and hepatic glucose metabolism; FGF21 modulates lipid metabolism and insulin sensitivity; the FGF23–α-Klotho–FGFR1 axis maintains phosphate and vitamin D homeostasis. These findings suggest FGFR as a potential therapeutic target for metabolic disorders [26,27].

#### 2.1.4. FGF-FGFR Interaction Networks and Regulatory Mechanisms

The complexity of the FGF–FGFR network arises not only from ligand–receptor binding diversity but also from multilayered regulation. HSPGs, serving as co-receptors for paracrine FGFs, enhance binding affinity and promote dimerization, thereby establishing local concentration gradients and shaping spatial signal distribution [23]. Endocrine FGFs depend on Klotho proteins to alter FGFR ligand selectivity, enabling precise regulation in specific tissues [42].

Beyond canonical ligand–receptor interactions, FGFRL1 acts as a molecular rheostat of FGF signaling. As a decoy receptor, FGFRL1 sequesters soluble FGF ligands and constrains the signaling range of select FGFs, thereby restricting signal amplitude and spatial spread. Under oncogenic conditions, dysregulated FGFRL1 expression or altered subcellular localization can reprogram FGF–FGFR signaling networks through context-dependent modulation of effector pathways such as PI3K–Akt and ERK1/2, potentially contributing to therapeutic resistance [43]. Thus, integrating FGFRL1 into comprehensive models of the FGF–FGFR network is imperative for understanding the dynamic regulation and plasticity of this signaling system.

Negative regulation is critical for maintaining spatiotemporal specificity: Sprouty (SPRY) proteins inhibit the GRB2/SOS–RAS axis to limit MAPK activation; SEF binds to receptor complexes to suppress kinase activity or promote endocytosis; dual-specificity phosphatases (DUSPs) terminate signaling by dephosphorylating MAPKs [44,45]. Furthermore, structural studies reveal that in the resting state, D1-acid box interactions maintain receptor autoinhibition; ligand binding induces receptor dimerization and conformational changes that relieve inhibition, thereby initiating downstream signaling [23,34].

### 2.2. Distribution Patterns of FGFR Aberrations in Solid Tumors

As a key receptor tyrosine kinase, FGFR exhibits structural and functional aberrations across multiple solid tumors. Large-scale genomic studies (e.g., TCGA, ICGC) have provided systematic evidence characterizing the frequency, types, and associations of FGFR aberrations with molecular subtypes, clinical features, and population differences.

#### 2.2.1. Statistical Analysis of FGFR Aberrations in Solid Tumors Based on Large Databases

In a sequencing study of 4853 solid tumors, the overall prevalence of FGFR aberrations was 7.1%, comprising 66% amplifications, 26% mutations, and 8% fusions [46]. This provides a reference for understanding the overall landscape of FGFR aberrations.

Notably, reported pan-cancer prevalence estimates are strongly influenced by assay design and analytical sensitivity. DNA-based hybrid-capture panels and whole-exome sequencing often under-detect FGFR rearrangements compared with RNA-based NGS or anchored multiplex PCR, which provide higher sensitivity for in-frame and rare-partner fusions [47]. In contrast, assays with higher copy-number resolution—such as high-depth targeted NGS or array-CGH—tend to detect more low-level FGFR amplifications [48,49]. Conversely, cfDNA-based NGS may miss subclonal or low-allele-fraction FGFR events because detection relies on tumor DNA fraction and shedding dynamics [50,51].

Pan-cancer analyses further reveal significant heterogeneity in FGFR isoform distribution across cancer types [52]. A combined TCGA and ICGC whole-genome study demonstrated that each cancer harbors an average of 4–5 driver mutations, with *FGFR* aberrations representing an important component that appears across multiple cancer types, though their oncogenic contribution varies by tumor type [53,54]. Additionally, a large-scale study of Chinese patient cohorts found an overall FGFR aberration rate of 7.0%, consistent with international data, while showing certain ethnic differences in cancer type distribution [55].

#### 2.2.2. Frequency and Type Distribution of FGFR Aberrations Across Cancer Types

*FGFR* aberrations exhibit marked tissue specificity and molecular preferences across tumor types. In urothelial carcinoma, *FGFR3* mutations are most prevalent, with an overall frequency of approximately 15–20%; detection rates reach 40–50% in non-muscle-invasive bladder cancer but decline to 10–15% in muscle-invasive or metastatic disease [56,57,58]. In intrahepatic cholangiocarcinoma (iCCA), *FGFR2* fusions are most common, occurring in approximately 10–16% of cases and correlating with distinct clinicopathological features [52,59]. Breast cancer harbors *FGFR* aberrations in 14–18% of cases, predominantly *FGFR1* amplifications [46,55]. Endometrial cancer frequently exhibits *FGFR2* mutations (e.g., S252W, N549K) in approximately 10–12% of cases, with established oncogenic activity [54,60]. While FGFR aberrations are generally rare in lung cancer, *FGFR1* amplification occurs in 15–20% of lung squamous cell carcinomas [46,61]. In glioblastoma, the overall prevalence of FGFR aberrations is approximately 2–8%, with *FGFR3* fusions being relatively common [62,63]. Additionally, certain rare tumors such as adenoid cystic carcinoma and ovarian malignant Brenner tumor display elevated proportions of FGFR fusions, typically associated with sensitivity to FGFR inhibitors [64].

In contrast to FGFR1–4, the role of FGFRL1 in cancer research remains underexplored. However, large-scale cancer genomics analyses together with functional studies in several tumor types have suggested its potential importance in solid tumors. On the one hand, transcriptomic and clinicopathological data indicate that FGFRL1 expression in prostate cancer, esophageal cancer, and small-cell lung cancer is associated with tumor progression and, in some cohorts, with patient outcome, suggesting biological roles that are at least partly distinct from those of classical FGFR1–4 [65,66,67]. On the other hand, in esophageal cancer, increased FGFRL1 expression correlates with disease progression and is directly repressed by miR-107, whereas in small-cell lung cancer FGFRL1 promotes chemoresistance via an ENO1–PI3K/Akt axis, and in pancreatic cancer models exosomal miR-210 from cancer stem cells targets FGFRL1 to drive macrophage M2 polarization and gemcitabine resistance [68,69]. These findings suggest significant variation in FGFR signaling dependency across tumors of different tissue origins.

#### 2.2.3. Association of FGFR Aberrations with Tumor Molecular Subtypes

*FGFR* aberrations display pronounced molecular subtype specificity across tumor types. In bladder cancer, *FGFR3* mutations predominantly occur in low-grade, non-muscle-invasive subtypes with papillary growth patterns and well-differentiated histology, suggesting they may define a distinct molecular subtype. Typical hotspot mutations (S249C, R248C, Y373C) involve extracellular cysteine substitutions that alter receptor conformation and confer ligand-independent activation [70,71,72]. In cholangiocarcinoma, *FGFR2* fusions and truncation variants (e.g., *FGFR2ΔE18*) represent mechanistically distinct driver events. *FGFR2* fusions typically act as solitary drivers activating RAS/MAPK and PI3K/AKT pathways to induce tumorigenesis, while *FGFR2ΔE18* causes receptor conformational stabilization and ligand-independent activation through structural deletion, indicating significant differences in signaling dependency and drug sensitivity [73,74]. In breast cancer, *FGFR1* amplification is enriched in hormone receptor-positive (ER+) subtypes, particularly Luminal B-like (high-proliferation HR+) tumors, and closely associates with endocrine therapy resistance and poor prognosis [75,76]. A large-scale functional study identified 71 newly confirmed oncogenic variants among 160 FGFR mutations; *FGFR2* mutations in endometrial cancer significantly correlate with molecular subtypes, showing markedly elevated frequency in microsatellite instability-high (MSI-H) tumors [77]. Table 1 summarizes the distribution patterns, frequency ranges, and molecular characteristics of FGFR aberrations in major solid tumors.

#### 2.2.4. Impact of Racial, Geographic, and Age Differences on FGFR Aberration Distribution

FGFR alterations also vary across populations. In a Chinese cohort study (n = 58), the frequency of FGFR alterations in colorectal cancer was 31%, significantly higher than in Western populations; in gastric cancer, the frequency was 16.8%, likewise indicating geographic variation. Regarding subtype distribution, *FGFR1* alterations were most common among Chinese patients (56.8%), followed by *FGFR3* (17.7%) and *FGFR2* (14.4%), differing from Western populations [55]. In Indian bladder cancer patients, the *FGFR3* mutation rate is 19%, comparable to the 15–30% range in European and American populations, but with differences in clinicopathological correlations [78]. Age also influences FGFR aberration distribution. In childhood and adolescent cancers, mutations, fusions, and amplifications of *FGFR1–4* are all relatively common, particularly in low-grade neuroepithelial tumors where they are significantly more prevalent than in adults, suggesting developmental stage-related differences in oncogenic mechanisms [79].

Taken together, available sequencing datasets indicate that the overall prevalence of FGFR alterations is broadly similar between Chinese and predominantly European/North American cohorts. By contrast, the relative contribution of individual tumor types differs by ethnicity and geography, with Chinese pan-cancer series enriched for FGFR-altered gastric and colorectal cancers, whereas Western datasets are more heavily weighted toward urothelial, breast, and endometrial tumors. Robust prevalence data from patients of African, South Asian, or Latin American ancestry remain scarce, underscoring the need for more ethnically diverse, globally representative genomic studies.

#### 2.2.5. Aberrant Alternative Splicing of FGFR1–3 in Solid Tumors

Alternative splicing represents an important yet underappreciated source of functional diversity within the FGFR family. For FGFR1–3, the best-characterized splicing events occur in the third immunoglobulin-like domain (IgIII), giving rise to the epithelial-associated IIIb and mesenchymal-associated IIIc isoforms, which differ in ligand specificity and downstream signaling output. This splicing program is tightly regulated during development but is frequently dysregulated in cancer, contributing to altered tumor–stromal interactions and oncogenic signaling plasticity [80].

In FGFR1, expression of truncated splice variants such as FGFR1β, which lacks the first Ig-like domain, enhances ligand binding affinity and mitogenic signaling and has been associated with aggressive behavior in breast cancer and other solid tumors [81]. In FGFR2, a shift from the IIIb to the IIIc isoform is recurrently observed during carcinoma progression and has been mechanistically linked to epithelial–mesenchymal transition and increased invasiveness, partly through aberrant regulation by splicing factors such as PTBP1 [82,83]. FGFR3 also exhibits cancer-associated splice variants beyond the canonical IIIb/IIIc forms, including variants that alter C-terminal regulation of kinase signaling, leading to enhanced oncogenic activity and altered therapeutic sensitivity in prostate cancer models [84].

Collectively, aberrant alternative splicing of FGFR1–3 adds an additional layer of complexity to FGFR dysregulation in solid tumors, with implications for tumor biology, biomarker interpretation, and therapeutic targeting.

## 3. Molecular Mechanisms and Pathophysiological Roles of FGFR Signaling Pathways

### 3.1. FGFR Downstream Signaling Network

As key members of the receptor tyrosine kinase family, FGFRs activate multiple signaling cascades through intracellular kinase domain autophosphorylation upon ligand binding, thereby regulating core biological processes including cell proliferation, differentiation, survival, and migration.

#### 3.1.1. Activation Mechanisms of Classical Downstream Pathways: MAPK/ERK, PI3K/AKT, and PLCγ

FGFRs primarily function through three classical pathways: RAS-MAPK/ERK, PI3K-AKT, and PLCγ-PKC. The coordinated activation of these pathways constitutes the core signaling network [3]. These three signaling axes are spatially and temporally interconnected, collectively determining cell fate transitions and response patterns (Figure 1).

##### RAS-MAPK/ERK Pathway Activation Mechanism

Upon FGFR activation, the receptor substrate FRS2 is rapidly phosphorylated and recruits the GRB2-SOS complex, triggering the RAS-RAF-MEK-ERK cascade. Activated ERK1/2 translocates to the nucleus and phosphorylates transcription factors including ELK1, FOS, and JUN, driving transcription of cell cycle-related genes [85]. Additionally, the ERK5 (*MAPK7*) branch can also be activated by FGFR signaling, mediating cell migration and stress responses [86]. Recent studies reveal that the FGFR-FRS2-GRB2 complex maintains signaling persistence after receptor endocytosis, indicating spatiotemporal-specific regulation [87].

##### PI3K-AKT Pathway Activation Mechanism

PI3K-AKT pathway activation also depends on FRS2. Phosphorylated FRS2 recruits GAB1 (GRB2-associated binder 1), providing binding sites for PI3K. Activated PI3K generates PIP3, which further recruits and activates AKT and PDK1 [88]. Full AKT activation requires both PDK1 and mTORC2. Activated AKT regulates cell survival (BAD, FoxO3a, p53), growth (TSC2-mTOR pathway), metabolism (GSK3β), and cell cycle through phosphorylation of multiple substrates, playing critical roles in tumor cell survival, proliferation, and metabolic reprogramming [89].

##### PLCγ-PKC Pathway Activation Mechanism

PLCγ pathway activation is relatively unique. PLCγ1 can directly bind to and be phosphorylated by specific sites on the FGFR activation loop, subsequently hydrolyzing PIP2 to generate IP3 and DAG. In FGFR1, this direct interaction is primarily mediated by autophosphorylation of the juxtamembrane tyrosine residue Y766, which serves as a canonical docking site for the SH2 domains of PLCγ1 and is essential for its recruitment and activation. IP3 promotes calcium release from the endoplasmic reticulum, elevating intracellular Ca^2+^; DAG and Ca^2+^ jointly activate PKC [19]. PKC further regulates cell differentiation, transcription, and membrane receptor function. In tumors, aberrant PKC activation is closely associated with invasion, angiogenesis, and drug resistance [90].

Overall, these three classical pathways are both independent and tightly interconnected. FGFR signaling maintains cellular homeostasis through spatial restriction, negative feedback, and pathway crosstalk. Its aberrant sustained activation not only promotes tumor cell proliferation and anti-apoptosis but also provides a molecular basis for drug resistance development.

#### 3.1.2. Crosstalk Between FGFR Signaling and Other Growth Factor Receptor Pathways

FGFR pathways exhibit extensive crosstalk with multiple RTKs, playing important roles in tumorigenesis and drug resistance. FGFR-EGFR interactions primarily depend on shared downstream adaptors (GRB2, GAB1) and common signaling nodes (RAS-MAPK, PI3K-AKT), generating synergistic oncogenic effects and mutual compensation in certain tumors, thereby affecting sensitivity to single-agent targeted therapy [91]. In non-small-cell lung cancer, co-occurrence of *FGFR1* amplification with *EGFR* alterations (amplification or mutations) has been observed in individual cases and small cohorts. The associated signaling is mediated through GRB2/GAB1 and may lead to downstream hyperactivation, though this remains a rare phenomenon overall, requiring larger studies to validate its clinical significance [92,93].

Mechanistic studies suggest that FGFRs can form complexes with EGFR and other RTKs at the membrane or in endosomes, or undergo transactivation, thereby altering signal output and promoting drug resistance. For example, studies report that FGFR4 can enhance EGFR oncogenic signaling and affect responses to EGFR inhibitors, suggesting potential targetable interaction mechanisms. However, the frequency and prevalence of such direct interactions remain unclear [94].

FGFR-MET interactions are also observed in triple-negative breast cancer, gastric cancer, and glioblastoma. Their co-expression or complementary activation is often associated with enhanced invasiveness and poor prognosis, with mechanisms frequently involving synergistic activation of the PI3K-AKT-mTOR axis, suggesting potential benefit from dual-target inhibition in selected patients [91,95].

In hormone receptor-positive breast cancer, FGFRs (especially FGFR2) exhibit functional interactions with estrogen receptor (ER): FGFR signaling promotes Erα phosphorylation and transcriptional activation via PI3K-AKT, and synergizes with EGFR/IGF-1R, thereby accelerating acquired endocrine resistance. This evidence indicates that comprehensive characterization of FGFR interactions with RTK/nuclear receptor networks holds significant value for optimizing combination or sequential therapy strategies and overcoming resistance [96].

#### 3.1.3. Spatiotemporal Dynamics and Feedback Mechanisms of Signal Transduction

Precise transmission of FGFR signals depends on complex negative feedback mechanisms and spatiotemporal regulatory systems to ensure effective control of signal specificity, intensity, and duration.

##### Multilevel Regulation by Negative Feedback Molecules

At the negative feedback level, the SPRY protein family serves as important inducible inhibitors of FGFR-mediated MAPK signaling. SPRY1–4 can be induced and translocate to the plasma membrane upon receptor stimulation, competitively blocking GRB2-SOS complex recruitment by binding GRB2, thereby inhibiting RAS activation. Additionally, SPRY can interfere at multiple levels of the MAPK axis, forming dual upstream/midstream inhibition to limit signal amplification. *DUSP6* (MKP-3), an ERK-specific dual-specificity phosphatase, rapidly terminates MAPK signaling by dephosphorylating activated ERK1/2. Its expression is positively regulated by ERK, forming a classical negative feedback loop that enables timely signal termination following rapid initiation [97,98,99]. Similar expression to FGF (SEF) inhibits FGFR activity through interaction with the receptor or receptor complexes and can competitively bind FRS2α to block downstream signal transmission. SEF can also alter signal output characteristics in different subcellular compartments by affecting receptor localization [44,100].

##### Precise Regulation of Receptor Endocytosis and Degradation

Receptor endocytosis and degradation represent another critical mechanism controlling signal intensity and duration. Ligand-activated FGFRs undergo clathrin-mediated endocytosis into endosomes, where they can either recycle to the plasma membrane for continued signaling or traffic to lysosomes for degradation. Receptor ubiquitination is mediated by specific E3 ubiquitin ligases (such as NEDD4/CBL), determining lysosomal sorting and affecting signal duration. Experimental evidence indicates that FGFR1 ubiquitination is not required for endocytosis but is essential for lysosomal trafficking and degradation; defective ubiquitination can increase receptor recycling and prolong signal transduction [101,102].

##### Spatiotemporal-Specific Signal Regulation Mechanisms

Regarding spatiotemporal dynamics, FGFR signaling exhibits significant spatial and temporal specificity: spatially, plasma membrane-localized receptor complexes are typically associated with rapid MAPK pathway activation, while endosomal receptor complexes more readily trigger metabolic pathways such as PI3K-AKT in certain cell types; temporally, FGFR-mediated signaling typically shows early MAPK dominance, followed by dynamic succession involving negative feedback inhibition and participation of other pathways [87,103]. This multilevel spatiotemporal regulation ensures orderly coordination of different biological processes according to temporal and spatial sequences.

#### 3.1.4. Molecular Mechanisms of Aberrant Pathway Activation in Tumorigenesis and Progression

Aberrant activation of FGFR signaling pathways is a critical driver in various malignancies, commonly achieved through gene amplification, activating point mutations, and chromosomal translocations/fusions, leading to sustained downstream signaling that promotes tumor proliferation, survival, invasion, and therapeutic resistance [52].

##### Pathway Hyperactivation by Gene Amplification

Gene amplification causes receptor overexpression and amplifies signal intensity. *FGFR1* amplification is relatively enriched in lung squamous cell carcinoma, with multiple reports indicating an incidence of approximately 15–20%. Amplified FGFR1 synergistically drives tumorigenesis and progression through sustained activation of RAS-MAPK (upregulating cell cycle regulators such as *CCND1*/*CCNE1*), PI3K-AKT-mTOR (promoting metabolic reprogramming and survival), and PLCγ-PKC (affecting cytoskeleton and migration) pathways [93,104,105]. FGFR2 amplification occurs at low overall frequencies in gastric cancer but is relatively prominent in certain diffuse-type or specific cohorts. It promotes epithelial–mesenchymal transition, invasion, and tumor growth by activating PI3K-AKT and MAPK pathways to upregulate epithelial–mesenchymal transition (EMT) transcription factors (such as Snail/Slug) and induce angiogenic factors (such as VEGF) [106].

##### Oncogenic Mechanisms of Activating Mutations

*FGFR3* mutations play important oncogenic roles in bladder cancer. Common mutations such as S249C, R248C, and Y373C, located in the receptor dimerization domain, cause constitutive receptor activation in the absence of ligand [107]. These mutations promote tumorigenesis through multiple mechanisms: sustained MAPK-ERK pathway activation phosphorylates and activates transcription factors ELK1 and AP-1, upregulating proliferation-related genes; meanwhile, aberrant PI3K-AKT pathway activation inhibits apoptosis by phosphorylating BAD and FoxO3a, enhancing tumor cell survival [108,109]. Activating *FGFR2* mutations commonly found in endometrial cancer (such as S252W, N549K) similarly enhance receptor kinase activity, thereby activating the PI3K-AKT-mTOR pathway to promote cell proliferation and metabolic reprogramming, while regulating cell cycle-related gene expression through the MAPK pathway [110].

##### Aberrant Activation Mechanisms of Fusion Genes

*FGFR2* fusions occur in approximately 10–16% of cholangiocarcinomas, with common fusion partners including *BICC1*, *AHCYL1*, and *PPHLN1* [111,112]. These fusion proteins typically retain the FGFR2 kinase domain but lose the ligand-binding domain and autoinhibitory region, resulting in constitutive kinase activity. Oncogenic mechanisms include sustained MAPK pathway activation upregulating oncogenes such as MYC and CCND1 to promote cell proliferation; PI3K-AKT-mTOR pathway promotion of protein synthesis and lipid metabolism; and STAT3 pathway activation to regulate the tumor microenvironment and immune evasion [113,114].

##### Pathway Reactivation Associated with Therapeutic Resistance

During FGFR inhibitor therapy, tumor cells can reactivate downstream signaling through multiple mechanisms to acquire resistance. Major mechanisms include the following: ① Secondary kinase domain mutations: In *FGFR2* fusion-positive patients, polyclonal mutations frequently emerge at progression, with hotspot sites including the “molecular brake” (N549/N550) and gatekeeper residue (V564), these alterations significantly reduce inhibitor binding and drive resistance; ② Bypass pathway activation: Compensatory activation of RTKs such as EGFR, MET, and IGF1R can maintain MAPK and PI3K-AKT signaling, thereby circumventing FGFR dependency; ③ Downstream molecular aberrations: Mutations or amplifications in *RAS*, *PI3K*, or *AKT* can directly activate key signaling axes, enabling tumor cells to escape inhibition [12,115].

##### Pathway Activation Through Tumor Microenvironment Modulation

Aberrant FGFR signaling activation also promotes tumor progression by remodeling the tumor microenvironment. On one hand, FGFR signaling upregulates angiogenic factors such as VEGF and PDGF, promoting neovascularization; on the other hand, FGFR signaling enhances expression of cytokines including TGF-β and IL-6, regulating cancer-associated fibroblast and immune cell functions to establish a pro-tumorigenic microenvironment. Additionally, FGFR pathways can promote extracellular matrix remodeling by upregulating matrix metalloproteinases (MMPs), thereby enhancing tumor invasion and metastatic potential [116,117,118,119].

Available phosphoproteomic and immunoblot analyses demonstrate that FGFR amplifications, activating mutations, and oncogenic fusions consistently engage MAPK/ERK and PI3K/AKT signaling relative to FGFR-wild-type controls [87,120,121]. However, quantitative studies across different isoforms, fusion partners, and cellular contexts show substantial variability in the magnitude of downstream activation, indicating that no uniform fold-change estimate can be applied across aberration classes or tumor types [122,123].

### 3.2. Interactions Between FGFR and the Tumor Microenvironment

#### 3.2.1. Regulatory Mechanisms of FGFR Signaling in Tumor Angiogenesis

Angiogenesis is a key feature of the tumor microenvironment, providing nutrients and oxygen for tumor growth and metastasis. FGFR signaling plays a central role in vessel formation and maintenance, with FGF2 being the prototypical pro-angiogenic factor. Upon binding to endothelial cell-surface FGFR1 and its co-receptor HSPG, FGF2 activates the PI3K-AKT and MAPK pathways, driving endothelial cell proliferation, migration, and lumen formation, and promotes basement membrane degradation and cellular infiltration by inducing MMPs [124]. This regulatory mechanism is detailed in (Figure 2). In the tumor microenvironment, FGF/FGFR signaling not only acts directly on endothelial cells but also indirectly promotes angiogenesis by modulating immune and stromal cells. For example, tumor-derived FGF2 enhances the survival and migration of tumor-associated macrophages (TAMs) and skews them toward an immunosuppressive phenotype, thereby establishing a microenvironment conducive to angiogenesis and tumor progression [125].

FGFR and VEGF signaling pathways exhibit significant complementarity and positive feedback regulation in tumor angiogenesis. FGFR activation can upregulate VEGF transcription and secretion, while VEGF signaling in turn enhances FGF expression, forming a bidirectional regulatory loop that cooperatively drives neovascularization under specific microenvironmental conditions (Figure 2). This crosstalk not only strengthens the angiogenic capacity of tumors but is also considered a key mechanism underlying compensatory resistance to anti-VEGF therapy. To disrupt such signaling interplay, multi-target tyrosine kinase inhibitors (TKIs), such as nintedanib, which simultaneously target VEGFR, FGFR, and PDGFR, have been developed to attenuate pathway cross-activation. Translational studies have shown that nintedanib effectively suppresses FGFR/VEGFR-mediated angiogenic signaling, improves tumor vascular structure and function, and enhances responsiveness to chemotherapy and immunotherapy. For instance, in non-small-cell lung cancer (NSCLC) xenograft models, nintedanib concurrently inhibits VEGFR and FGFR activities, significantly reduces tumor angiogenesis, and, when combined with immunotherapy, promotes CD8^+^ T cell infiltration, thereby augmenting antitumor immune responses [126,127,128].

It is noteworthy that aberrant FGFR activation not only Increases the number of blood vessels but also induces structural disorganization and elevated vascular permeability, which together promote tumor cell dissemination and affect drug delivery. Based on the concept of “vascular normalization,” moderate inhibition of FGFR signaling can help restore vascular architecture and enhance the efficacy of chemotherapy and immunotherapy [129,130]. In addition to FGF2, ligands such as FGF9 also play roles in microcirculatory regulation, influencing metastatic potential and tumor dynamics, further underscoring the multifaceted role of FGFR signaling in angiogenesis [131].

#### 3.2.2. Relationship Between FGFR and Immune Cell Infiltration and Immune Evasion

FGFR signaling not only regulates angiogenesis but also reshapes the immune microenvironment to influence immune infiltration and immune evasion. The underlying mechanisms involve cancer-associated fibroblasts (CAFs), myeloid cells, and cytokine networks, which collectively alter immune cell recruitment and function, ultimately affecting the efficacy of immune checkpoint inhibitors (ICIs) [132].

Different FGFR subtypes exert distinct immunomodulatory effects. In gastric cancer, FGFR1 expression correlates positively with CD8^+^, CD4^+^, and dendritic cell infiltration, whereas FGFR4 expression shows a negative correlation with lymphocyte infiltration. Functional studies reveal that FGFR1 overexpression enhances the response to PD-1 blockade, while FGFR4 overexpression weakens it, suggesting that different FGFR subtypes shape immune lineages through distinct mechanisms [133].

Mechanistic studies indicate that FGFR inhibition can remodel the tumor microenvironment to enhance effector T cell infiltration. In triple-negative breast cancer models, FGFR inhibition alters CAF secretory profiles and matrix remodeling, markedly increasing CD8^+^ T cell infiltration and suppressing tumor growth, supporting the rationale for FGFR-TKI and ICI combination therapy [134]. Additionally, FGFR inhibition reduces infiltration of myeloid-derived suppressor cells (MDSCs), decreases the incidence of distant metastases, and further potentiates antitumor immune responses [135].

A key mechanism involves the IFN-γ → JAK/STAT1/IRF1 axis. FGFR activation broadly attenuates IFN-γ-induced transcriptional programs, whereas FGFR inhibition restores them and enhances ICI responsiveness [136]. Tumor-type-specific datasets support this conserved inhibitory effect: in renal cell carcinoma and multiple solid-tumor models, constitutive FGFR signaling diminishes IFN-γ-induced STAT1 phosphorylation and downstream genes such as IRF1, CXCL10, B2M, and PD-L1, while FGFR blockade reinstates this pathway [136]. Similar suppression of interferon-stimulated genes by FGF/FGFR activation has been documented across cellular systems, suggesting a shared biological mechanism rather than a model-restricted phenomenon [137]. In bladder cancer, *FGFR3* activation further intersects with IFN-γ signaling by modulating PD-L1 stability through NEDD4-dependent ubiquitination or by altering IFN-γ-induced PD-L1 turnover in *FGFR3-TACC3* fusion models [138,139]. By contrast, evidence from FGFR2-fusion intrahepatic cholangiocarcinoma is limited and sometimes inconsistent, underscoring the need for tumor-specific interpretation of FGFR–interferon crosstalk [140,141].

On the other hand, in hepatocellular carcinoma, FGFR4 inhibition promotes PD-L1 ubiquitin–proteasome degradation via GSK3β activation and suppresses IL-2–induced STAT5 phosphorylation to limit Treg differentiation, thereby reducing PD-L1 levels and immunosuppression and enhancing anti-PD-1 efficacy [142]. These mechanisms are not contradictory but rather context-specific, depending on tumor type, FGFR subtype, and molecular background. This highlights the importance of considering tumor-specific contexts when interpreting FGFR–ICI interactions.

#### 3.2.3. Role of FGFR in Cancer-Associated Fibroblasts and Its Crosstalk with Tumor Cells

CAFs are the most abundant stromal cells in the tumor microenvironment and profoundly influence tumor progression by secreting growth factors, cytokines, and extracellular matrix (ECM) components. FGF/FGFR signaling plays a central role in CAF activation and functional maintenance, forming an interactive network among CAFs, immune cells, and tumor cells [143,144].

In gastric cancer, CAF-derived IGFBP7 enhances the FGF2/FGFR1–PI3K/AKT pathway, promoting TAM recruitment and polarization, thereby establishing a pro-tumorigenic CAF–TAM–tumor cell axis [145]. In addition, FGF2 sustains CAF activation via autocrine and paracrine mechanisms. Activated CAFs upregulate MMPs and their regulators (such as TIMPs) through FGFR-related pathways, driving ECM remodeling that both creates invasion channels for tumor dissemination and releases matrix-sequestered pro-tumorigenic factors to amplify signaling [146,147].

The heterogeneity of CAFs has garnered considerable attention. Single-cell sequencing and functional analyses reveal transcriptional and functional differences among CAF subtypes. Differential expression patterns of FGFR isoforms have been observed in some contexts, though such “CAF subtype–FGFR” correspondence is not universal and requires validation across tumor types [148,149].

Recent spatial and single-cell transcriptomic studies further contextualize this heterogeneity. In pancreatic ductal adenocarcinoma (PDAC), inflammatory CAF (iCAF), myofibroblastic CAF (myCAF), and antigen-presenting CAF (apCAF) programs occupy discrete spatial niches and show characteristic proximity to tumor or immune compartments, as demonstrated by integrated spatial and single-cell profiling [150,151]. Similar integrated approaches in breast, colorectal, lung, and skin cancers have delineated conserved CAF subclasses (e.g., iCAF- and myCAF-like populations) while revealing tumor-specific spatial organization and neighborhood interactions. For instance, in breast cancer, spatial transcriptomics combined with scRNA-seq identifies CAF subpopulations with distinct proximity to immune infiltrates and tumor cells, implicating them in immune evasion, tumor progression, and potentially modulating responses to immunotherapy [152]. In colorectal cancer, spatially resolved profiling uncovers multicellular immune hubs involving CAFs that organize immunosuppressive niches [153]. In non-small-cell lung cancer, similar analyses underscore the plasticity of stromal and myeloid compartments, while in cutaneous squamous cell carcinoma and melanoma, CAF-associated stromal remodeling at the tumor interface supports the existence of conserved yet context-dependent stromal states [154,155,156]. Collectively, these integrated datasets refine CAF subset annotation and strengthen evidence for their specialized roles within distinct tumor microenvironments.

FGFR inhibition can partially reverse the pro-tumorigenic effects of CAFs. In triple-negative breast cancer, FGFR blockade remodels CAF phenotypes, enhances CD8^+^ T cell infiltration, and boosts immune responses [134]. In lung adenocarcinoma, inhibition of the FGF/FGFR pathway suppresses CAF activation and reduces secretion of proinflammatory and prometastatic factors, thereby restraining tumor growth [157]. Furthermore, in prostate cancer, downregulation of miR-15/16 in CAFs relieves repression of the FGF2/FGFR1 axis, further promoting tumor growth and metastasis, highlighting the molecular complexity of CAF–tumor cell interactions [158].

#### 3.2.4. Molecular Mechanisms of FGFR Signaling in Tumor Invasion and Metastasis

FGFR signaling plays a pivotal role in multiple stages of tumor invasion and metastasis by regulating cell adhesion, motility, angiogenesis, and the immune microenvironment [159]. Among these, EMT is a key mechanism that endows tumor cells with migratory and invasive capabilities. Numerous reviews and experimental studies have demonstrated that FGF/FGFR signaling can initiate or potentiate EMT-related molecular programs, typically accompanied by altered expression of EMT-associated transcription factors (such as Snail, Slug, and Twist) and epithelial markers (such as E-cadherin), thereby promoting mesenchymal phenotypes and stem-like properties in tumor cells [130,160]. This transition not only enhances cellular motility and invasiveness but also confers stemness and therapeutic resistance.

In addition, FGFR signaling promotes ECM degradation by regulating the expression and activity of MMPs. Activated FGFR upregulates MMP-2, MMP-9, and MMP-14 via the MAPK and PI3K-AKT pathways. These proteases degrade key ECM components such as collagen, fibronectin, and laminin, thus creating invasion corridors for tumor dissemination. Meanwhile, MMPs can also activate latent growth factors and cytokines, further amplifying pro-metastatic signaling cascades [161].

During hematogenous metastasis, tumor cells must traverse the vascular endothelial barrier. FGFR signaling facilitates this process by regulating endothelial permeability and the expression of adhesion molecules, thereby promoting tumor cell intravasation and extravasation [162]. Studies have shown that FGF2 increases the permeability of endothelial tight junctions and upregulates ICAM-1 and VCAM-1 expression, providing favorable conditions for circulating tumor cell adhesion and transendothelial migration [163,164].

Furthermore, FGFR signaling plays a crucial role in the formation and maintenance of distant metastatic niches. In colorectal cancer liver metastasis, tumor-derived FGF19 acts on FGFR4 expressed in hepatic stellate cells, activating the JAK2–STAT3 pathway and inducing their differentiation into iCAFs. These iCAFs secrete proinflammatory mediators such as IL-1, complement C5a, and various chemokines, which recruit neutrophils and promote the formation of neutrophil extracellular traps (NETs). NETs capture circulating tumor cells and release growth-promoting factors, enhancing tumor cell adhesion, survival, and proliferation in the liver. This “soil conditioning” mechanism enables metastatic cells to adapt and thrive within the new tissue microenvironment (Figure 3) [165,166].

Consistent with these mechanistic insights, preclinical CRC liver metastasis models provide causal evidence supporting this FGF19–FGFR4–iCAF–NET axis. Genetic silencing of *FGF19* or pharmacologic inhibition of FGFR4 or JAK2 markedly reduces iCAF polarization and hepatic NET deposition [165,167]. Similarly, neutrophil depletion or NET disruption—such as DNase I administration or the use of peptidylarginine deiminase 4 (*Padi4*)-deficient mice—significantly suppresses metastatic outgrowth. Functionally, NETs not only trap circulating tumor cells but also deliver proliferative and metabolic cues that facilitate micrometastatic expansion, in line with broader evidence implicating NETs in distant seeding and reactivation of dormant cancer cells [167,168,169].

In addition, FGFR signaling regulates the chemokine system; for instance, it upregulates CXCR4 expression, thereby enhancing tumor cell chemotaxis toward CXCL12. In metastatic breast cancer, blockade of CXCR4 signaling not only directly suppresses metastasis but also increases T cell infiltration and improves responses to immunotherapy [170,171]. These findings suggest that the FGFR–CXCR4 signaling axis may represent a promising therapeutic target for anti-metastatic interventions.

While other signaling axes such as HGF/MET and CXCL12/CXCR4 are also critical for metastatic niche formation, FGFR signaling exhibits distinct yet cooperative functional characteristics. The HGF/MET pathway is classically recognized as a primary initiator of “invasive growth” and EMT driven by stromal paracrine signaling, whereas the CXCL12/CXCR4 axis predominantly functions as a “chemotactic compass” guiding tumor cell homing to specific organs like the bone and lung [172,173]. In contrast to single-pathway chemotactic axes such as CXCR4, FGFR signaling operates as both an invasion driver and a microenvironmental “soil remodeler.” FGFs can recruit neutrophils and reshape innate immune composition [174], while FGF2 potently modulates endothelial function and angiogenic programs, including VEGF induction, thereby facilitating intravasation and metastatic seeding [175,176]. FGFR activity also intersects with chemokine-driven trafficking, as FGF2 mobilizes stromal precursors that are subsequently recruited through the CXCL12–CXCR4 axis [177]. In parallel, extensive RTK crosstalk—particularly with MET—reinforces survival signaling and contributes to adaptive resistance to targeted therapies [178,179]. Collectively, these immune, vascular, and cooperative RTK effects define FGFR-driven niches as integrative hubs that strengthen metastatic establishment.

### 3.3. Phenotypic Diversity Driven by Aberrant FGFR Activation

Aberrant activation of the FGFR signaling pathway is a major driver of tumorigenesis and progression across various solid tumors. Distinct types of FGFR alterations exhibit significant differences at the molecular, cellular, and metabolic levels, leading to diverse tumor phenotypes. This has been well supported by large-scale sequencing analyses and comprehensive reviews [53,180].

#### 3.3.1. Distinct Oncogenic Mechanisms of Different Types of FGFR Alterations

Point mutations often cause constitutive receptor activation by altering receptor conformation or promoting aberrant dimerization. In urothelial carcinoma of the bladder, recurrent *FGFR3* hotspot mutations such as R248C, S249C and G370C cluster within the extracellular Ig-like domains or in the linker between the IgII and IgIII domains [181]. Biochemical analyses of the corresponding germline mutations that cause thanatophoric dysplasia type I have shown that substitution of these residues by cysteine introduces unpaired cysteines in the IgII–IgIII linker or juxtamembrane region, thereby promoting the formation of abnormal intermolecular disulfide bonds and covalent receptor dimers with ligand-independent kinase activation [182]. Together with crystallographic and biophysical studies that define how the D1–acid box module and HSPG–mediated FGF–FGFR assembly normally enforce an autoinhibited, ligand-dependent state [20,34], these data support a model in which Cys-substituting *FGFR3* mutations found in both bladder cancer and skeletal dysplasias drive constitutive signaling by stabilizing a covalent homodimer that at least partially bypasses D1–acid box-mediated autoinhibition and the strict requirement for HSPG-bridged dimerization, resulting in sustained trans-phosphorylation of the intracellular kinase domains.

The protein levels of FGFR3 are also associated with mutation status and histological differentiation [181,183]. In contrast, in endometrial carcinoma, *FGFR2* kinase-domain mutations (such as N550K and K660E) directly enhance kinase activity and alter receptor conformation. Structural studies indicate that these mutations often occur within the ATP-binding pocket or activation loop, disrupting the “molecular brake” mechanism and maintaining constitutive activation [184,185].

Gene amplification primarily enhances physiological signaling via receptor overexpression. Among such alterations, *FGFR1* amplification is most prominent in breast cancer. It increases cellular sensitivity to FGF, alters signaling strength and spatiotemporal dynamics, and frequently co-occurs with amplification of the 8p11–12 region harboring multiple oncogenic drivers, thereby influencing prognosis. *FGFR1* amplification is also strongly correlated with endocrine therapy resistance. Notably, copy number gain does not always correspond to protein overexpression across tumor subtypes, suggesting a critical role for post-transcriptional regulation in modulating receptor activity [75,96,186]. This genomic instability further contributes to the complexity of amplification-driven oncogenesis.

Gene fusions remodel receptor function through domain rearrangement. In cholangiocarcinoma, the recurrent FGFR2–BICC1 fusion retains the kinase domain but lacks the autoinhibitory region, leading to constitutive activation and altered subcellular localization. This fusion represents a key molecular rationale for FGFR2-targeted therapy.

Emerging evidence indicates that the oncogenicity of *FGFR2* fusion proteins is shaped not only by the preserved kinase domain but also by fusion-partner-encoded modules that govern oligomerization and intracellular trafficking [187]. Many recurrent partners, including *TACC3* and *BICC1*, provide coiled-coil, Sterile Alpha Motif, or related multimerization motifs that drive ligand-independent dimerization or higher-order clustering, thereby stabilizing constitutive FGFR activation [188]. For example, in *FGFR3–TACC3*, the TACC3 coiled-coil domain promotes aberrant oligomerization and redirects the fusion protein to mitotic spindle and centrosomal structures, producing ectopic signaling outputs and mitotic defects [189]. Similarly, BICC1 contains domains that facilitate spontaneous multimerization and influence membrane–endosomal or centrosomal localization, ultimately altering receptor trafficking dynamics [190]. These trafficking alterations—shifting the balance between recycling and lysosomal degradation, or redistributing receptors across plasma membrane, endosomal, or pericentrosomal compartments—can prolong kinase activity and reshape downstream signaling amplitude and duration. Consistent observations across engineered cell systems and functional genomic studies support these mechanisms and highlight disruption of fusion-driven oligomerization or trafficking as a potential therapeutic strategy complementary to kinase inhibition.

Another prototypical example is the FGFR3–TACC3 fusion, which was first identified and remains most prevalent in glioblastoma, with an incidence of approximately 2–5%. Beyond driving constitutive activation, its fusion partner TACC3 regulates cytoskeletal organization, spindle assembly, and cell division, thereby promoting malignant phenotypes. Moreover, this fusion induces distinct transcriptional and metabolic programs depending on the histological and cellular context [191,192,193].

Taken together, different types of FGFR alterations define distinct oncogenic mechanisms and downstream signaling networks: point mutations activate receptors through conformational changes or aberrant dimerization, amplifications enhance signal amplitude via a dosage effect, and fusions induce structural and localization remodeling. This molecular diversity underlies the heterogeneity of clinical phenotypes and therapeutic responses, highlighting the necessity of tailoring FGFR-targeted strategies according to specific alteration types and accompanying genomic contexts.

#### 3.3.2. The Role of FGFR Alterations in Cancer Stem Cell Maintenance and Differentiation

FGFR signaling plays a central role in the self-renewal and pluripotency maintenance of cancer stem cells (CSCs). In glioblastoma, for instance, FGFR1 sustains CSC stemness and EMT-related phenotypes through an ADAMDEC1–FGF2 autocrine loop that activates ERK1/2 signaling and upregulates transcription factors such as ZEB1 [194]. Moreover, in other solid tumors, such as non-small-cell lung cancer with FGFR1 amplification, the same alteration has been shown to induce “stem cell-like” characteristics, suggesting that this pathway exerts a conserved role in maintaining stemness across different tissue contexts [195].

This regulation extends beyond transcriptional control and involves chromatin remodeling mechanisms. Nuclear-localized FGFR1 can bind chromatin and colocalize with active histone marks, as supported by ChIP-seq and transcriptomic analyses, thereby reorganizing the transcriptional program of stemness-associated genes and influencing therapeutic sensitivity [196].

In hematologic malignancies—particularly myeloid or mixed-phenotype leukemias harboring *FGFR1* fusions—the constitutive kinase activity of fusion proteins interferes with myeloid differentiation, leading to differentiation blockade, accumulation of immature progenitors, and clonal expansion that confers enhanced chemoresistance [197,198,199]. Patient-derived models and single-cell sequencing analyses have further demonstrated the heterogeneity in differentiation states driven by FGFR alterations. *FGFR1* amplification or overexpression often results in disrupted differentiation hierarchies: some cells retain CSC-like features, while others display aberrant differentiation-related transcriptional profiles. The coexistence of these subpopulations is closely associated with treatment response variability [200,201].

The CSC niche, composed of CAFs, vascular endothelial cells, and other stromal components, sustains CSC activity through the secretion of exogenous FGFs [119,202]. In colorectal and other solid tumor models, stromal cell-derived or membrane-bound FGFs (such as FGF2) activate downstream FGFR signaling to promote tumor cell migration, invasion, and stem-like phenotypes. Recent studies have further demonstrated that FGFR signaling cooperates with multiple downstream pathways, including MAPK, PI3K, STATs, and SRC family kinases, to orchestrate these processes [203,204]. Furthermore, CSCs often localize within specialized microenvironmental niches, such as perivascular regions or tumor-invasive fronts, where local FGFR overexpression correlates positively with CSC enrichment. This suggests a spatially coordinated interaction between FGFR signaling and microenvironmental cues in shaping CSC heterogeneity and site-specific behavior [205,206].

Overall, FGFR abnormalities maintain CSC phenotypes and shape differentiation trajectories through multilayered mechanisms—including transcriptional regulation, chromatin remodeling, disruption of differentiation programs, and microenvironmental interactions—collectively driving tumor resistance, recurrence, and invasion. However, the precise molecular consequences vary significantly among tumor types and distinct *FGFR* alteration forms.

#### 3.3.3. Effects of FGFR Signaling on Tumor Metabolic Reprogramming

FGFR signaling plays a pivotal role in tumor metabolic reprogramming, particularly in glycolytic and lipid metabolic pathways. In lung squamous cell carcinoma models, FGFR1 signaling activates the PI3K–AKT–mTOR axis, leading to HIF-1α accumulation and upregulation of GLUT1, thereby enhancing glucose uptake and glycolysis—the hallmark Warburg effect. In addition, FGFR1 regulates glucose metabolism through its interaction with PKM2, modulating its phosphorylation status and subcellular localization. Metabolomic analyses have further revealed that FGFR aberrations promote glycolytic bias by altering lactate dehydrogenase (LDH) isoenzyme composition—upregulating LDHA while downregulating LDHB—which collectively supports tumor growth and metabolic flexibility [207,208,209,210].

In terms of lipid metabolism, FGFR signaling has been shown to activate lipogenic pathways across multiple tumor models, particularly in *FGFR3*-altered bladder cancer. It upregulates SREBP1 downstream targets, such as SCD1 and FASN, thereby promoting monounsaturated fatty acid synthesis and membrane biogenesis. These findings indicate that FGFR aberrations can drive lipid metabolic reprogramming. Lipidomic studies have also demonstrated a close association between the RTK/PI3K signaling axis and lipid metabolism, whereby FGFR–PI3K signaling supports downstream transduction by modulating levels of phospholipid precursors, such as PIP_2_, ultimately influencing membrane composition and signal propagation [211,212,213].

#### 3.3.4. Single-Cell Heterogeneity of FGFR Expression and Its Biological Significance

Single-cell sequencing has revealed a high degree of heterogeneity in FGFR expression within tumor tissues. Multiple single-cell and spatial transcriptomic studies have shown that FGFR expression varies markedly across cellular subpopulations, and FGFR-high cells are frequently colocalized with invasive, stem-like phenotypes or microenvironmental niches enriched in angiogenic and immunosuppressive features. This phenomenon has been observed, to varying extents, across several solid tumors, including pancreatic cancer, head and neck squamous cell carcinoma, and lung cancer [214,215,216].

Pseudotime trajectory and clonal dynamics analyses further demonstrate that FGFR expression correlates with cell differentiation states and clonal selection. For example, in head and neck squamous cell carcinoma and breast cancer, FGFR-high subpopulations typically exhibit transcriptional signatures resembling stem/progenitor cells, whereas FGFR-low or -negative subgroups show a trend toward differentiation lineages. Under therapeutic pressure, rare resistant or “refuge” subclones may expand and drive tumor recurrence—a dynamic process directly observed in longitudinal studies of FGFR-driven cholangiocarcinoma [13,217].

In addition, single-cell ligand–receptor interaction analyses and spatial transcriptomic tools have been applied to dissect the roles of FGFR-mediated signaling circuits among tumor cells, CAFs, and immune cells. These integrative analyses provide fresh insights into therapeutic response and resistance mechanisms and suggest that FGFR-targeted therapies should account for cooperative and compensatory interactions within the tumor microenvironment [218].

## 4. FGFR-Targeted Therapy: Drug Development and Clinical Applications

### 4.1. Classification and Mechanisms of FGFR Inhibitors

The development of FGFR inhibitors has evolved from non-selective multi-target inhibition toward highly selective and precise targeting. With deeper insights into FGFR structural biology and advances in rational drug design, a variety of strategies have emerged, including small-molecule TKIs, monoclonal antibodies, ADCs and novel delivery platforms. Understanding the design principles and clinical profiles of different inhibitor classes is essential for precision oncology and the development of next-generation therapeutics.

#### 4.1.1. Design Concepts and Clinical Performance of Selective vs. Non-Selective Inhibitors

Early FGFR inhibitors were predominantly non-selective multi-target TKIs, such as nintedanib and dovitinib. Nintedanib inhibits VEGFR, FGFR, and PDGFR simultaneously and demonstrated moderate efficacy in NSCLC but was frequently associated with diarrhea, hepatotoxicity, and vascular-related adverse events [128,219]. Dovitinib exhibits inhibitory activity against FGFR1–3, VEGFR, FLT3, and c-KIT, yet clinical trials in breast and renal cancers reported limited efficacy and substantial toxicity, restricting its clinical utility [220,221]. These early attempts highlighted a key trade-off: while non-selective inhibitors benefit from broad target coverage and potential multi-target synergism, their lack of specificity often leads to off-target toxicity and suboptimal therapeutic precision.

With advances in structural biology, the development of selective FGFR inhibitors has become increasingly sophisticated. These agents are designed to exploit the unique structural features of FGFR kinase domains, thereby improving target specificity and minimizing off-target effects [222]. Erdafitinib, the first selective FGFR inhibitor approved by the U.S. FDA, demonstrates high affinity toward FGFR1–4 (IC_50_: 1.8–10.8 nM). In patients with *FGFR3*-altered urothelial carcinoma, erdafitinib achieved an objective response rate (ORR) of 40%, markedly outperforming chemotherapy, which yielded an ORR of 13% [5,223].

Pemigatinib, a reversible ATP-competitive inhibitor, binds to the ATP-binding pocket of FGFR1–3, effectively suppressing their kinase activities. In the FIGHT-207 basket trial, pemigatinib demonstrated an ORR of 26.5% and a median progression-free survival (PFS) of 4.5 months among patients harboring FGFR fusions or rearrangements [224].

Futibatinib, a next-generation covalent FGFR inhibitor, exhibits potent antitumor activity, particularly in tumors harboring *FGFR2* and *FGFR3* mutations. Through optimized scaffold design, futibatinib overcomes several resistance-associated mutations and shows strong promise in both preclinical and early-phase clinical studies. Its molecular configuration enhances binding affinity for FGFR kinases while reducing nonspecific inhibition of other kinases. This highly selective targeting of FGFR2 and FGFR3 translates into superior efficacy and improved tolerability in genetically defined tumor subtypes [225].

#### 4.1.2. Development Strategies for Monoclonal Antibodies and Antibody–Drug Conjugates

Monoclonal antibodies (mAbs) inhibit FGFR signaling by blocking ligand–receptor interactions or preventing receptor dimerization, representing a mechanism distinct from that of small-molecule TKIs. Bemarituzumab is a humanized monoclonal antibody targeting FGFR2b, which inhibits ligand binding of FGF7 and FGF10. In a phase II clinical trial involving patients with gastric and gastroesophageal junction cancer, bemarituzumab in combination with chemotherapy significantly improved progression-free survival (PFS) (9.5 vs. 7.4 months) and demonstrated a favorable efficacy profile [226].

Vofatamab (B-701) is a monoclonal antibody targeting FGFR3, which acts through allosteric binding to the extracellular domain to block receptor activation. In early clinical studies, vofatamab combined with pembrolizumab in patients with metastatic urothelial carcinoma refractory to platinum-based therapy showed acceptable safety and preliminary signals of efficacy, suggesting that this combination approach warrants further clinical investigation [227].

ADCs combine the target specificity of monoclonal antibodies with the cytotoxic potency of chemotherapeutic payloads, earning them the nickname “biological missiles”. However, the development of FGFR-targeted ADCs faces multiple challenges, including the optimization of antibody selection, linker–payload design, and management of FGFR-specific toxicities [228].

Aprutumab ixadotin (BAY 1187982) was among the first FGFR2-targeting ADCs. Although it demonstrated selective cytotoxic activity against FGFR2-overexpressing tumors in preclinical models, its phase I clinical trial was terminated due to severe proteinuria and hematologic toxicities. Subsequent reviews have cited this case as an example to illustrate linker-associated toxicity and the unique risk profile of FGFR-targeted ADCs [229,230,231]. A novel tetravalent anti-FGFR1 ADC has recently been developed to enhance binding avidity and internalization efficiency. In vitro studies showed 5–10-fold greater cytotoxic potency compared to traditional bivalent ADCs, and superior antitumor efficacy was further validated in xenograft models [232,233].

The design of the ADC linker is a critical determinant of both therapeutic efficacy and safety. Cleavable linkers, such as the valine–citrulline linker, release their cytotoxic payload upon lysosomal enzymatic cleavage, while non-cleavable linkers require complete degradation of the antibody to liberate the drug [234]. Most FGFR-targeted ADCs employ cleavable linkers to ensure efficient intracellular drug release within target cells [235]. Emerging pH-sensitive or glutathione-responsive linkers exploit the unique biochemical characteristics of the tumor microenvironment to achieve precise site-specific drug release, potentially improving the therapeutic index and mitigating the impact of FGFR expression heterogeneity on treatment outcomes [236,237].

#### 4.1.3. Advances in Novel Targeted Delivery Systems and Nanomedicine Carriers

The efficient internalization properties of FGFR confer a unique advantage for developing targeted delivery systems [187]. Upon ligand binding, FGFR undergoes clathrin-mediated endocytosis, a process that has been leveraged to engineer diverse FGFR-targeted carriers. For instance, constructs that use FGF1 as the recognition module and are conjugated to fluorescent oligomers can integrate targeting, tracking, and toxin loading. A representative molecule, 3xGFPp_FGF1–MMAE, exhibits nanomolar affinity and marked cytotoxicity toward FGFR1-high cells in vitro, while showing minimal effects on FGFR1-negative cells, indicating potential for imaging-guided and precision delivery applications [238].

Nanotechnology provides a platform for improving the stability, targetability, and therapeutic index of FGFR inhibitors, including liposomes, polymeric nanoparticles, and inorganic nanocarriers, but still faces challenges in clinical translation such as Enhanced Permeability and Retention effect (EPR) heterogeneity, in vivo clearance, and scaled-up manufacturing [239,240]. Notably, the robust EPR effect observed in rapidly growing murine xenografts is substantially weaker and more heterogeneous in human solid tumors [241]. Clinical and preclinical studies using quantitative imaging of radiolabeled or MRI/PET-tracked nanomedicines indicate that, in patients, only a limited and highly variable fraction of the injected dose accumulates in tumor tissue, with variability influenced by tumor type, anatomical site, stromal architecture, vascular normalization status, and prior therapies [242,243]. This mismatch between preclinical and clinical EPR behavior is thought to be one important factor contributing to the observation that many EPR-dependent nanomedicines with impressive efficacy in mouse models have achieved only modest, if any, overall survival benefit in late-phase clinical trials [244,245].

To mitigate this translational gap, current strategies increasingly explore functional “EPR imaging” (e.g., dynamic contrast-enhanced MRI or nanocarrier-based MRI/PET) for patient and lesion stratification, transient modulation of tumor vasculature in rational combinations to improve perfusion, and the development of actively targeted or stimuli-responsive carriers designed to be less reliant on passive EPR for intratumoral delivery [246,247,248,249].

Targeted liposomes typically achieve active targeting through surface modification with anti-FGFR antibodies or ligands, enabling prolonged circulation time and accumulation in tumor tissues. Preclinical studies have demonstrated that encapsulating FGFR-active compounds in targeted liposomes or other nanocarriers can achieve superior antitumor efficacy compared to free drug and a more controllable systemic toxicity profile in animal models [250,251].

Polymer–drug conjugates (PDCs) offer an additional option for delivering FGFR inhibitors. By tethering drugs to polymers via biodegradable linkages, PDCs enable controlled release and targeted delivery. PDC formulations of FGFR inhibitors have shown improved pharmacokinetic profiles and enhanced antitumor activity in preclinical and early-phase studies; however, further evidence is required to establish their clinical utility [252,253,254].

In addition, smart stimuli-responsive delivery systems leverage features of the tumor microenvironment to achieve precise release. For example, pH-responsive nanocarriers accelerate drug release under acidic conditions, whereas enzyme-responsive carriers rely on tumor-overexpressed proteases such as MMPs to cleave linkers and liberate the payload. These “microenvironment-responsive” strategies have demonstrated promising controlled-release behavior and targeting potential in vitro and in selected in vivo models, yet a meaningful gap to clinical translation remains [255,256,257].

### 4.2. Clinical Efficacy Evaluation of Approved FGFR Inhibitors

The evidence-based evaluation of approved FGFR inhibitors provides a scientific foundation for precision oncology strategies. Table 2 summarizes the key trial designs and efficacy outcomes of major FGFR inhibitors across different cancer types. With the regulatory approvals of pemigatinib, erdafitinib, and futibatinib in multiple countries, the body of clinical data regarding their efficacy and safety has expanded, while real-world studies continue to supplement findings from pivotal registration trials.

#### 4.2.1. Clinical Trial Outcomes of Pemigatinib in Cholangiocarcinoma

The FIGHT-202 trial (Phase II, multicenter, open-label) demonstrated the antitumor efficacy of pemigatinib in previously treated patients with *FGFR2* fusion or rearrangement–positive cholangiocarcinoma. The initial analysis reported an ORR of approximately 35.5%, a disease control rate (DCR) of about 82%, a median PFS of 6.9 months, and a median overall survival (OS) of 21.1 months [4]. An independent review committee (IRC)-based long-term follow-up analysis further validated these results, showing an ORR of 37.0%, median duration of response (DOR) of 9.1 months, median PFS of 7.0 months, and median OS of 17.5 months [258].

In the broader FIGHT-207 trial (Phase II, basket study encompassing multiple tumor types and FGFR alterations), the ORR was 26.5% among patients with FGFR fusions or rearrangements, while those harboring activating non-kinase domain mutations exhibited an ORR of only 9.4%, underscoring a marked difference in pemigatinib sensitivity across distinct *FGFR* alteration types, and providing direct evidence to support molecular stratification in clinical practice [224].

The ongoing FIGHT-302 study (Phase III, randomized controlled trial; pemigatinib vs. gemcitabine + cisplatin, 2:1 randomization; primary endpoint: PFS) is designed to evaluate the efficacy and safety of pemigatinib as first-line therapy for advanced *FGFR2*-rearranged cholangiocarcinoma. The trial has not yet reported final endpoint data [259].

In terms of safety, both clinical trials and subsequent pooled analyses have identified hyperphosphatemia as the most common treatment-related adverse event, with a relatively high incidence rate. Effective management—including dose adjustment, low-phosphate diets, and use of phosphate binders—is recommended to maintain treatment adherence and optimize long-term benefit. Real-world data generally confirm efficacy and safety outcomes consistent with clinical trial findings but highlight the critical role of toxicity management in improving clinical outcomes [4,258].

#### 4.2.2. Efficacy and Safety of Erdafitinib in Bladder Cancer

The BLC2001 trial, a pivotal phase II registration study, evaluated erdafitinib in 99 patients with advanced urothelial carcinoma harboring FGFR alterations. The trial reported an ORR of 40%, a median DOR of 5.6 months, and a median PFS of 5.5 months. Subgroup analysis revealed a higher ORR among patients with *FGFR3* mutations (49%) compared with those with FGFR fusions (33%), indicating that the type of FGFR alteration may influence therapeutic response [5,260].

The phase III THOR trial, which enrolled 266 patients with FGFR-altered advanced urothelial carcinoma, compared erdafitinib with standard chemotherapy. Erdafitinib significantly improved OS, achieving a median OS of 12.1 months versus 7.8 months with chemotherapy (*p* = 0.005), thereby establishing its role as the standard of care for patients with FGFR-altered urothelial carcinoma [261].

The RAGNAR trial, an international, single-arm phase II study, was designed to systematically evaluate the efficacy and safety of erdafitinib in patients with advanced or metastatic solid tumors harboring *FGFR1–4* alterations. In the overall study population, erdafitinib achieved an ORR of 29% and a median DOR of 6.9 months, supporting the tumor-agnostic druggability of FGFR-driven malignancies. Notably, in the cholangiocarcinoma cohort, erdafitinib demonstrated robust antitumor activity, with an ORR of approximately 60% and a median PFS of around 8.5 months [262].

With respect to safety, pooled analyses identified the most common treatment-related adverse events as dry skin (74%), fatigue (36%), diarrhea (32%), and hyperphosphatemia (>50%) [265]. Hyperphosphatemia represents a mechanism-related, on-target toxicity, while ocular adverse events—including central serous retinopathy—occurred in about 25% of patients, were largely reversible, and required regular ophthalmologic evaluation and serum phosphate monitoring to optimize the balance between efficacy and tolerability [266].

#### 4.2.3. Clinical Advances of Futibatinib and Next-Generation FGFR Inhibitors

The FOENIX CCA2 trial, a pivotal phase II registration study, enrolled 103 patients with advanced iCCA harboring *FGFR2* fusions or rearrangements. The study reported an ORR of 42%, a median PFS of 9.0 months, and a median OS of 21.7 months, demonstrating durable and clinically meaningful benefit in this molecularly defined population [191].

Exploratory genomic analyses indicated no significant efficacy difference between *BICC1* and non *BICC1* fusion partners (ORR = 41.7% vs. 44.6%). Moreover, a rapid decline in circulating tumor DNA (ctDNA) after treatment correlated with improved clinical outcomes, underscoring the potential utility of liquid biopsy for response monitoring and early detection of resistance. Mechanistic studies revealed that secondary *FGFR2* kinase domain mutations represent the predominant mechanism of acquired resistance, with specific sites such as L617 reducing inhibitor sensitivity. The irreversible binding mode of futibatinib confers activity against certain resistant mutants; however, polyclonal secondary mutations remain a major therapeutic challenge [267].

Among emerging agents, RLY 4008 (lirafugratinib), a highly selective FGFR2 inhibitor, has demonstrated exceptional target specificity and broad activity against resistance mutations in early clinical evaluation. Preliminary results from the phase I ReFocus study showed encouraging antitumor activity at the recommended dose in *FGFR2* driven malignancies. These findings suggest that sequential use of more selective FGFR inhibitors may offer a novel strategy to extend the durability of FGFR targeted therapy [263,264].

#### 4.2.4. Real World Evidence on the Efficacy and Safety of FGFR Inhibitors

A large-scale pharmacovigilance analysis based on the FDA Adverse Event Reporting System (FAERS) evaluated real world safety profiles of erdafitinib, pemigatinib, and futibatinib. Overall, the safety spectrum in real world settings was consistent with findings from registration trials, although certain rare but serious adverse events appeared at slightly higher frequencies, likely reflecting patient heterogeneity and reporting bias [268]. For instance, pemigatinib associated calciphylaxis has been reported in real world cases but was not observed in clinical trials [269].

In terms of clinical effectiveness, real world outcomes generally aligned with pivotal trial data but demonstrated modest variations due to differences in patient selection and response assessment. In a real-world cohort of 120 pemigatinib treated patients, the real-world ORR was 59.2%, and the median real world PFS was 7.4 months, exceeding those reported in the FIGHT 202 trial [270]. This improvement may reflect selection bias toward patients with more favorable baseline characteristics and non-uniform response assessment methods.

Real world populations also exhibited greater clinical complexity, including a higher prevalence of hepatic dysfunction and heavily pretreated disease, which likely contributed to outcome variability and underscored the limitations of real-world evidence studies [258,271]. Furthermore, inconsistent quality of FGFR genomic testing markedly influenced efficacy evaluations. Patients assessed using standardized next-generation sequencing (NGS) platforms showed significantly better outcomes than those tested by non-standardized methods [73]. Thus, establishing uniform FGFR testing standards is essential for enhancing clinical reproducibility and improving the reliability of real-world evidence.

## 5. Mechanisms of Resistance and Therapeutic Strategies

### 5.1. Molecular Basis of Resistance to FGFR Targeted Therapy

Resistance to FGFR targeted therapy represents a major barrier to sustaining durable clinical benefit, arising from highly diverse and complex molecular mechanisms. With the expanding clinical use of FGFR inhibitors, therapeutic resistance has become increasingly apparent. Detailed elucidation of the underlying mechanisms provides crucial insights into treatment failure and informs the design of next-generation inhibitors and predictive biomarkers [52]. Current evidence suggests that FGFR targeted resistance primarily results from acquired FGFR structural mutations, activation of compensatory bypass pathways, and clonal evolution driven by tumor heterogeneity, reflecting the multifaceted nature of therapeutic escape.

#### 5.1.1. Molecular Characteristics and Structural Basis of Acquired Resistance Mutations

Acquired mutations constitute the most prevalent mechanism of resistance to FGFR inhibitors, with gatekeeper mutations being the archetypal example. These mutations occur within the ATP binding pocket of the kinase domain, where substitution of small residues (e.g., valine) with bulkier amino acids (e.g., methionine or phenylalanine) introduces steric hindrance that impedes inhibitor access or proper orientation, thereby diminishing drug affinity. Canonical examples include *FGFR1* V561M, *FGFR2* V564F, and *FGFR3* V555M [272,273,274]. Structural studies reveal that such mutations disrupt key hydrophobic interactions, promote stabilization of the active kinase conformation, and may enhance catalytic turnover, while also altering the kinetic stability of the inhibitor–protein complex rather than merely increasing ATP affinity [275].

Beyond gatekeeper substitutions, additional secondary mutations within the FGFR kinase domain have been implicated in clinical resistance. In patients with *FGFR2* fusion–positive cholangiocarcinoma, approximately 60% develop one or more secondary *FGFR2* kinase domain alterations upon progression, most frequently at N550 and V565 [12]. These mutations exhibit heterogeneous modes of action: N550 variants disrupt hydrogen bond networks within the hinge/molecular brake region, destabilizing the autoinhibited conformation, whereas gatekeeper like mutations primarily confer resistance via steric interference with inhibitor binding [276,277].

Molecular dynamics simulations further demonstrate that V564F/V565F mutations markedly reduce the structural stability of complexes formed between reversible ATP competitive inhibitors (e.g., pemigatinib and infigratinib) and FGFR2, providing a mechanistic explanation for the distinct resistance profiles observed across clinical and preclinical settings [275,278].

#### 5.1.2. Bypass Signaling Activation as a Mechanism of Resistance

Upon pharmacologic blockade of FGFR signaling, tumor cells frequently exploit alternative RTKs and their downstream cascades to sustain proliferation and survival. Multiple RTKs—including EGFR, MET, ErbB3, and EphB3—have been validated across diverse preclinical models as key mediators of acquired resistance to FGFR inhibitors [279,280,281]. Their activation is driven by multifactorial mechanisms, encompassing transcriptional upregulation, enhanced protein stability, and increased availability of extracellular ligands, collectively enabling tumor cells to circumvent FGFR dependency and restore oncogenic signaling.

PI3K–AKT signaling represents a pivotal downstream effector of FGFR and constitutes a major bypass mechanism underlying therapeutic resistance. Loss or functional impairment of PTEN results in sustained accumulation of PIP3, thereby maintaining AKT activation even under FGFR inhibition and consequently diminishing inhibitor sensitivity [282,283,284]. In addition, reduced expression or inactivation of pleckstrin homology-like domain family A member 1 (PHLDA1)—a critical negative regulator of PI3K–AKT signaling—can unleash AKT activity and promote a resistant phenotype, a mechanism functionally validated in multiple RTK-driven tumor models [285,286].

Reactivation of the RAS–MAPK cascade represents another prevalent route to escape FGFR-targeted therapy. Acquired NRAS/KRAS amplifications or gain-of-function mutations have been detected in resistant tumor specimens, while downregulation of DUSP6—a phosphatase specifically targeting ERK—can sustain ERK phosphorylation and propagate proliferative output. Such convergent rewiring of the MAPK pathway highlights the inherent vulnerability of monotherapy against a single RTK, which is frequently bypassed by resilient downstream oncogenic signaling [287,288].

Beyond canonical oncogenic cascades, metabolic rewiring and autophagy have emerged as increasingly critical drivers of therapeutic resistance. In gastric cancer models, FGFR inhibition has been shown to activate the TAK1/AMPK axis and enhance autophagic flux, thereby enabling resistant cells to withstand metabolic and pharmacologic stress; concurrently, transcription factor EB (TFEB), a lysosome–autophagy master regulator, undergoes nuclear translocation to upregulate lysosomal and autophagy-associated gene programs, further strengthening cellular stress tolerance [289,290]. These findings collectively illustrate that, when classical RTK pathways are blocked, tumor cells can acquire alternative survival routes through metabolic adaptability and autophagy activation, ultimately inducing or sustaining resistance. However, substantial heterogeneity has been observed across distinct tumor types and molecular subgroups, suggesting that rational combinations or sequential strategies co-targeting downstream MAPK/PI3K signaling or metabolic vulnerabilities may outperform FGFR monotherapy in delaying or overcoming resistance.

#### 5.1.3. The Role of Tumor Heterogeneity and Clonal Evolution in Resistance

Tumor heterogeneity represents a fundamental driver of resistance to FGFR-targeted therapies, manifesting across both temporal and spatial dimensions. Multi-omics profiling and rapid autopsy analyses have demonstrated that FGFR inhibitors can profoundly reshape the clonal architecture of cholangiocarcinoma: dominant FGFR-altered clones prior to treatment diminish following FGFR blockade, whereas minor subclones harboring secondary resistance mutations expand under therapeutic pressure and ultimately gain dominance [115,217]. Single-cell and high-resolution omics studies further reveal that these resistant subpopulations may pre-exist at low frequency before therapy, with drug exposure promoting their accelerated expansion. Such competitive or replacement clonal dynamics ultimately select for fitter clones, which govern the emergence of resistant phenotypes [291].

ctDNA has emerged as a powerful tool for real-time tracking of clonal evolution. Longitudinal analyses demonstrate that resistance-associated mutations can arise prior to radiographic progression, indicating opportunities for earlier therapeutic intervention. Moreover, ctDNA profiling reveals the prevalent nature of polyclonal resistance: diverse resistant mutations may coexist within the same patient, underscoring the tumor’s ability to deploy multiple escape strategies simultaneously. This complexity explains why next-generation inhibitors alone often fail to fully eradicate resistance and highlights the necessity of rational combination therapies [12,292].

Clinical and multi-omics investigations suggest that resistant and sensitive clones may engage in competitive, coexistent, or even cooperative interactions. However, current evidence largely stems from observational molecular profiling, with limited real-time tumor ecology or functional validation. Future studies integrating high-resolution longitudinal omics, dynamic ctDNA monitoring, and ecological modeling will be essential to elucidate the evolutionary trajectories of resistance and optimize personalized therapeutic strategies [115].

#### 5.1.4. Epigenetic Alterations Shaping Sensitivity to FGFR-Targeted Therapy

Epigenetic dysregulation plays a multilayered role in modulating resistance to FGFR-targeted therapies. Aberrations in chromatin-remodeling complexes—particularly loss of function of the SWI/SNF complex—represent an important molecular dimension influencing therapeutic response and have been directly implicated in adaptive resistance to FGFR inhibition in selected models [293]. In lung cancer and other solid tumors, deficiency of key SWI/SNF subunits (such as SMARCA4/BRG1) profoundly reorganizes chromatin architecture and transcriptional circuitry, thereby reshaping cellular sensitivity to multiple anticancer agents.

Mechanistically, SMARCA4 loss promotes heterochromatin accumulation and elevates replication stress, rendering tumor cells increasingly dependent on ATR signaling and consequently more vulnerable to ATR inhibitors in vitro and in vivo [294,295]. Moreover, SMARCA4/2 deficiency drives enhancer reprogramming and global transcriptional rewiring, activating MYC-linked or lineage plasticity programs and perturbing the DNA damage response (DDR), which in certain contexts confers potential susceptibility to PARP inhibition or ATR/PARP combinatorial strategies [296,297]. In addition, SMARCA4 defects attenuate expression of apoptosis-related programs—including mechanisms associated with impaired ER-to-mitochondria Ca^2+^ transfer—resulting in diminished responses to chemotherapy and other targeted agents (Figure 4) [298]. Although these molecular axes provide strong biological rationale for exploiting drug vulnerabilities in SMARCA4-deficient contexts, whether SMARCA4 directly drives clinical resistance to FGFR inhibitors remains to be validated through larger multi-omics datasets and longitudinal cohorts.

Abnormalities in histone deacetylases (HDACs) may also contribute to resistance against FGFR-targeted therapies. Emerging evidence suggests that members such as HDAC2 promote tumor cell survival and drug-refractory phenotypes by modulating cell-cycle regulators, apoptosis pathways, EMT and cancer stemness features [299,300]. Although these findings have been supported in vitro and in animal models, causal evidence within the specific context of FGFR inhibition remains limited, underscoring the need for further functional and translational studies to define their actionable relevance.

DNA methylation represents another critical epigenetic determinant. FGFR-altered tumors frequently demonstrate unique methylation landscapes associated with transcriptional states and therapeutic responsiveness; however, systematic evidence remains insufficient, particularly regarding whether FGFR inhibitor treatment induces widespread remodeling of methylation programs involving DNA repair and cell death genes [301]. In glioblastoma, *MGMT* promoter methylation is strongly associated with enhanced sensitivity to alkylating chemotherapy, suggesting that impaired DNA repair capacity may modulate treatment response. Yet, whether similar principles apply to FGFR inhibition remains hypothetical, lacking direct clinical validation [302,303].

Non-coding RNAs, particularly microRNAs (miRNAs), play pivotal roles in therapeutic resistance regulation. Accumulating evidence demonstrates that the miR-100 family orchestrates FGFR3 expression through direct targeting of its 3′ untranslated region (3′UTR). In urothelial carcinoma, reduced miR-100 expression correlates with aberrant FGFR3 activation and enhanced downstream signaling, implicating the miR-100-associated non-coding RNA regulatory networks in modulating drug sensitivity and acquired resistance in FGFR-driven malignancies [304,305].

Emerging insights further reveal that lysosomal biology constitutes an underappreciated mechanism of resistance to small-molecule targeted therapies. Under therapeutic stress, the TFEB undergoes nuclear translocation and induces the expression of lysosomal and autophagy-related genes, thereby augmenting lysosomal biogenesis and functional capacity [290]. Certain small-molecule inhibitors—particularly lipophilic weak bases—undergo lysosomal sequestration, which substantially diminishes their effective intracellular concentrations at target sites. Moreover, alterations in vacuolar-type H^+^-ATPase (V-ATPase) function or expression modulate lysosomal acidification status, potentiating this resistance phenotype [306,307,308]. While these mechanisms have been validated across diverse experimental models and provide a compelling framework for understanding intracellular drug distribution and resistance, direct clinical evidence specific to FGFR inhibitors remains limited, warranting rigorous investigation.

### 5.2. Prediction, Monitoring, and Early Intervention Strategies for Therapeutic Resistance

As FGFR-targeted therapies gain widespread clinical adoption, acquired resistance has emerged as the principal barrier limiting durable patient benefit. Establishing a systematic framework for resistance management necessitates not only mechanistic elucidation of resistance pathways but also the development of robust predictive tools, implementation of dynamic surveillance systems, and formulation of evidence-based early intervention protocols. The convergence of multi-omics-integrated prediction models, real-time ctDNA monitoring platforms, and artificial intelligence-driven personalized treatment algorithms has ushered in unprecedented opportunities for comprehensive resistance management in FGFR-driven malignancies. This multidimensional approach to resistance mitigation holds promise not only for enhancing therapeutic efficacy but also for postponing resistance emergence, ultimately improving patient clinical outcomes and long-term survival.

#### 5.2.1. Resistance Prediction Models Integrating FGFR Genomic Alterations and Proteomic Profiling

Integrative prediction models incorporating FGFR genomic alterations and proteomic profiling have emerged as a pivotal approach for precision resistance prognostication. Multi-omics frameworks integrating genomic and proteomic signatures have demonstrated superior predictive performance relative to single-biomarker approaches across multiple cohort studies. However, prospective clinical validation remains nascent, with existing evidence predominantly derived from retrospective cohort analyses and exploratory early-phase trial investigations [309,310].

Proteomics and phosphoproteomics investigations have illuminated complex nonlinear relationships between genetic alterations and functional phenotypes, providing a rational foundation for patient-specific therapeutic target prioritization [311]. In cholangiocarcinoma, patients harboring identical *FGFR2* fusions frequently exhibit marked heterogeneity in molecular profiles and downstream signaling cascades. Converging evidence indicates that aberrant activation of compensatory pathways closely associates with intrinsic or acquired resistance, manifesting distinct patient-specific patterns [191,287]. Consequently, elucidating resistance mechanisms typically necessitates serial biopsies or longitudinal ctDNA profiling to guide subsequent targeted strategies or sequential treatment selection. Recent investigations and comprehensive reviews consistently support the clinical utility of ctDNA in detecting resistance clones and informing therapeutic decision-making in *FGFR2* fusion-positive iCCA [312,313,314].

Beyond driver mutations and signaling pathway activation, genomic instability may also influence therapeutic efficacy. In certain solid tumors, homologous recombination deficiency correlates with treatment response; however, evidence supporting its role as a predictive biomarker for FGFR therapeutic efficacy remains insufficient. High tumor mutational burden (TMB) is typically accompanied by multi-pathway abnormalities and complex signaling networks, which may increase resistance risk. The relationship between TMB and FGFR inhibitor efficacy requires further validation through prospective studies [13,52].

Identification and functional validation of proteomic biomarkers are essential for establishing reliable prediction models. Large-scale screening of various FGFR-aberrant tumors has identified multiple candidate proteins associated with resistance [311,315]. For instance, low PHLDA1 expression is closely associated with resistance in RTK-driven tumors, including models related to the FGFR pathway. PHLDA1 negatively regulates AKT signaling; its loss leads to excessive PI3K-AKT activation, thereby promoting resistance [286,316]. These findings highlight the important value of proteomics in elucidating resistance mechanisms and constructing prediction models, laying the foundation for developing individualized resistance prediction tools based on functional parameters such as protein expression and phosphorylation status.

#### 5.2.2. The Value of ctDNA Dynamic Monitoring of FGFR Secondary Mutations in Early Resistance Detection

ctDNA, as a core tool of liquid biopsy, has demonstrated unique advantages in monitoring *FGFR* secondary mutations. Compared with traditional tissue biopsy, ctDNA detection is characterized by its non-invasive, real-time, and repeatable nature, making it particularly suitable for dynamic assessment during treatment. In cholangiocarcinoma, ultra-high-sensitivity digital PCR technology has achieved detection of *FGFR2* secondary mutations at frequencies as low as 0.1% [115]. However, its application still faces several challenges, including the risk of false negatives when tumor burden is low or ctDNA shedding is limited, as well as difficulties in distinguishing tumor-derived mutations from clonal hematopoiesis-associated mutations. Additionally, the relatively short half-life of ctDNA (approximately 1.5–2 h) imposes stringent requirements on the timeliness of sample collection and processing [317,318,319].

Clinical studies have validated the practical value of ctDNA dynamic monitoring. Prospective investigations of *FGFR2* fusion-positive cholangiocarcinoma have established standardized protocols: baseline detection followed by initial assessment at 4 weeks of treatment, with subsequent monitoring every 8 weeks. Results demonstrate that the detection of emergent *FGFR2* secondary mutations in ctDNA precedes radiographic progression by an average of 10–14 weeks, providing a critical window for early intervention [292]. Furthermore, ctDNA analysis has revealed the polyclonal nature of resistance: individual patients may harbor multiple mutations, including *V564F*, *N549H*, and *E565A*, at different timepoints, indicating that tumors evade therapy through multiple pathways. This complexity suggests that single next-generation inhibitors are unlikely to fully overcome resistance, and combination therapies may offer superior efficacy [115].

ctDNA and imaging assessment demonstrate clear complementarity. In targeted therapy for urothelial carcinoma, ctDNA can reflect molecular-level resistance or response earlier, while CT/MRI retains irreplaceable advantages in tumor burden quantification and local control assessment [13]. Therefore, their integration can form a stratified management strategy: when ctDNA indicates resistance but imaging shows no progression, follow-up intervals should be shortened and biopsy considered; when both indicate progression, treatment strategies should be adjusted promptly. However, large-scale systematic studies to validate this combined monitoring approach are lacking, and future research should focus on this area.

#### 5.2.3. Application Prospects of Artificial Intelligence in FGFR Resistance Prediction and Individualized Decision-Making

Artificial intelligence (AI) has demonstrated significant potential in predicting targeted drug resistance. By integrating genomic, transcriptomic, proteomic, and clinical data, researchers have constructed various machine learning models to assess resistance risk. In multiple pan-cancer studies, algorithms such as random forest and Bayesian multi-kernel learning have significantly improved prediction accuracy by integrating genetic features including point mutations, copy number variations, and gene fusions, along with clinical variables such as age, sex, stage, and prior treatment history, providing a methodological foundation for FGFR-related resistance risk assessment [320,321,322]. Deep learning has further advanced model performance, convolutional neural networks (CNNs) can identify complex genomic variation patterns. In cholangiocarcinoma research, CNN-based models have revealed multiple copy number and structural variation features associated with FGFR2 resistance [191,323].

Multimodal data fusion is considered a key development direction for AI in FGFR resistance prediction. By integrating ctDNA, imaging, pathology, and clinical information, complex interactions between molecular features and clinical phenotypes can be captured. Multiple studies have shown that multimodal models outperform single-modality models in precision medicine tasks [324,325]. For example, Vanguri et al. integrated radiomics, pathomics, and genomic features to predict immunotherapy response in 247 patients with advanced NSCLC, achieving an AUC of 0.80, significantly superior to single-modality models [326].

However, multimodal AI research targeting FGFR aberrations remains in early stages, predominantly consisting of small-sample or single-center explorations that lack external validation and multi-center consistency assessment [327,328]. Additionally, cross-modal data differences in dimensionality, distribution, and noise characteristics, along with data missingness and quality heterogeneity, as well as issues of model interpretability and reproducibility, all limit clinical translation. Future research requires interdisciplinary collaboration to establish standardized data collection and annotation systems, promote large-scale multi-center studies, and incorporate dynamic ctDNA monitoring information to enhance the verifiability and clinical applicability of FGFR resistance prediction models.

### 5.3. Therapeutic Strategies for Overcoming Resistance

The emergence of FGFR-targeted therapy resistance represents a significant challenge in clinical practice, necessitating innovative therapeutic strategies to overcome this obstacle. Strategies for reversing resistance must be based not only on in-depth understanding of resistance mechanisms, but also require the development of next-generation therapeutic tools and combination treatment regimens through multidisciplinary collaboration. From next-generation FGFR inhibitors designed at the molecular structural level, to exploration of synergistic mechanisms in immunotherapy combination strategies, and to formulation of precision treatment regimens based on individual resistance profiles, these innovative solutions offer new hope and avenues for overcoming FGFR-targeted therapy resistance. With deepening understanding of resistance mechanisms and continued advances in therapeutic technologies, individualized resistance reversal strategies will become an important development direction for FGFR-targeted therapy.

#### 5.3.1. Design Principles and Preclinical Research of Next-Generation FGFR Inhibitors

The core objective of next-generation FGFR inhibitors is to overcome acquired resistance mutations, particularly the steric barriers imposed by gatekeeper mutations. Irreversible inhibitors exemplified by futibatinib can achieve sustained inhibition by forming covalent bonds with cysteine residues in the FGFR kinase domain, thereby overcoming gatekeeper and other acquired mutations to a certain extent [225]. Structural studies indicate that utilizing electrophilic warheads such as acrylamide to form covalent linkages with cysteine can enhance inhibitory activity against gatekeeper mutations and prolong duration of action; however, the specific binding mode, selectivity, and pharmacokinetic effects remain dependent on molecular design [329].

The development of multi-targeted inhibitors aims to block bypass resistance. AZD4547 is a multi-targeted TKI primarily inhibiting FGFR1/2/3, with additional activity against VEGFR2/VEGFR3 and CSF1R. It can directly block FGFR-driven proliferative signaling in tumor cells while possessing anti-angiogenic potential through inhibition of FGFR1/2 and VEGFR2/3 signaling in endothelial cells. Consequently, it demonstrates potential advantages in tumors with active angiogenesis and/or synergistic FGF/VEGF signaling [130]. Another direction involves combined inhibition of FGFR and cell cycle pathways. Since FGFR signaling regulates the cell cycle through cyclin D1, preclinical studies have shown that in NSCLC models with FGF3/4/19/CCND1 amplification, LY2874455 and the CDK4/6 inhibitor abemaciclib exhibit synergistic effects and can reverse resistance; in estrogen receptor-positive breast cancer, both theoretical rationale and early clinical studies support the validity of triple combination therapy targeting ER, CDK4/6, and FGFR [330,331].

Novel drug delivery strategies also provide expanded avenues for FGFR-targeted therapy. Anti-FGFR3 antibody–drug conjugates (such as LY3076226) have demonstrated tolerable dose ranges and preliminary antitumor activity in phase I clinical studies, but require larger-scale validation [332]. Additionally, nanoparticle or lipid carriers (such as LNPs, PLGA nanoparticles, and liposomes) can significantly enhance tumor tissue accumulation of FGFR inhibitors and improve the therapeutic index in preclinical studies, suggesting their potential in drug delivery optimization; however, their clinical translation remains dependent on further evaluation of pharmacokinetics and toxicity profiles [333,334].

#### 5.3.2. Synergistic Potential and Mechanistic Basis of FGFR Inhibition and Immunotherapy

FGFR signaling plays a central regulatory role in shaping the tumor immune microenvironment. Beyond promoting tumor cell proliferation and survival, aberrant FGFR activation can induce expression of immunosuppressive factors through multiple pathways, enhance the activity of regulatory T cells (Tregs) and myeloid suppressor cells, thereby attenuating antitumor immune responses [132,335]. Multiple murine model studies have demonstrated that FGFR inhibitors can partially reverse the immunosuppressive state, manifested by increased CD8^+^ T cell infiltration and decreased proportions of immunosuppressive cells, thereby creating a more favorable immune environment for the action of ICIs [134,336].

Based on this rationale, the combination of FGFR inhibitors with PD-1/PD-L1 checkpoint inhibitors is considered to have potential synergistic effects. However, this effect exhibits significant tumor type and molecular dependency. For example, erdafitinib can reduce PD-L1 expression on tumor cells and enhance T cell activation in certain models, thereby improving the efficacy of anti-PD-1 antibodies; conversely, in FGFR3-activated bladder cancer models, FGFR3 inhibition can upregulate PD-L1 levels through modulation of the E3 ubiquitin ligase NEDD4, paradoxically attenuating immune responses [138,139,337]. These discrepancies suggest that complex bidirectional regulatory mechanisms exist between the FGFR pathway and immune checkpoint signaling, and combination strategies should be optimized in conjunction with specific molecular contexts and tumor ecosystem features.

In recent studies, FGFR has also been explored as a novel target for cell-based therapy. CAR-T cells targeting FGFR4 have demonstrated significant antitumor activity in multiple solid tumor models, providing feasibility evidence for FGFR-targeted cellular immunotherapy [338,339]. In bladder cancer, FGFR3 can regulate PD-L1 ubiquitination and stability through phosphorylation of NEDD4, thereby affecting CD8^+^ T cell function [72,138]. Furthermore, preclinical evidence indicates that combination of FGFR inhibitors with CAR-T or ICIs can enhance therapeutic efficacy through dual mechanisms: on one hand, suppressing immunosuppressive signaling, and on the other hand, improving T cell infiltration and effector function. However, direct preclinical or clinical evidence remains limited at present, and further validation in standardized models and multi-center studies is needed in the future [340].

#### 5.3.3. Formulation of Individualized Treatment Strategies Based on Resistance Mechanisms

The core of personalized therapy lies in the precise identification of patients’ molecular characteristics. Different types of FGFR aberrations exhibit differential drug sensitivity: some studies suggest that patients with FGFR gene amplification or overexpression are more sensitive to ATP-competitive inhibitors, whereas those with fusions or specific activating mutations demonstrate more heterogeneous responses [52]. Monitoring of resistance mutation spectra is particularly critical. For patients harboring gatekeeper mutations, irreversible FGFR inhibitors may be effective [191], whereas if resistance mechanisms involve bypass pathway activation, a combination of FGFR inhibitors with pathway-specific inhibitors may hold greater promise [341]. Through ctDNA monitoring technology, the emergence of resistance mutations can be tracked in real time, providing evidence for timely adjustment of treatment regimens.

Beyond molecular characteristics, phenotype-based individualized strategies are equally important. In highly vascularized FGFR-aberrant tumors, FGFR inhibitors combined with anti-angiogenic agents have demonstrated synergistic effects in preclinical models and warrant further exploration in certain solid tumors [132,342]; in tumors with high FGFR4/KLB expression, selective FGFR4 inhibitors have shown feasibility in early studies, and preliminary results of their combination with PD-1 inhibitors suggest potential benefits [343]. Immune phenotype also serves as a key reference: in immunologically “hot” tumors, combination of FGFR inhibitors with checkpoint inhibitors may produce more significant therapeutic efficacy; whereas in immunologically “cold” tumors, it may be necessary to first remodel the microenvironment through FGFR inhibitors or other strategies before introducing immunotherapy [222].

Furthermore, dynamic and adaptive treatment strategies further expand the scope of individualized management. In early treatment phases, the most effective FGFR inhibitors can be prioritized to achieve rapid disease control; when resistance signals emerge, combination or sequential therapy should be promptly implemented. Through intermittent dosing or dose adjustment, selection pressure on resistant clones can be reduced, thereby delaying resistance development. In prostate cancer androgen deprivation therapy, intermittent treatment has been proven to delay resistance and improve quality of life, and similar strategies warrant exploration in FGFR-targeted therapy [344,345,346].

## 6. Combination Treatment Strategies and Multidisciplinary Integration

### 6.1. Combination Modalities of FGFR Inhibitors with Conventional Therapies

As an emerging targeted therapy, FGFR inhibitors have demonstrated application potential in various solid tumors. However, their single-agent efficacy is often limited by resistance and tumor heterogeneity. Therefore, combining them with conventional treatments (chemotherapy, radiotherapy, and surgery) has become a critical direction for optimizing existing therapeutic paradigms. This section focuses on exploring combination modalities of FGFR inhibitors with conventional therapies, encompassing temporal sequencing, dose optimization, mechanisms of action, clinical validation, and toxicity management. By integrating the latest clinical and preclinical data, we evaluate their contribution to the therapeutic paradigm of solid tumors.

#### 6.1.1. Studies on Temporal Sequencing and Dose Optimization in Combination with Chemotherapy

In cholangiocarcinoma, the combination of FGFR inhibitors (represented by pemigatinib) with chemotherapy (such as gemcitabine, GEM) has emerged as an important exploratory direction. Preclinical studies have demonstrated that GEM can induce downstream pathway activation (including phosphorylation of FRS2 and ERK) in certain FGF-activated tumor models, accompanied by alterations in cell cycle and mitotic markers; in cell lines and xenograft models harboring *FGFR2* fusions, the combination of GEM with pemigatinib produced synergistic antitumor effects both in vitro and in vivo, significantly enhancing apoptosis through inhibition of FGFR-mediated survival signaling, suggesting potential clinical value for concurrent combination in chemotherapy-naive patients with FGF-activated backgrounds [347]. Given that other receptor tyrosine kinases have shown schedule-dependent effects, systematic comparison of three strategies—“targeted therapy followed by chemotherapy,” “chemotherapy followed by targeted therapy,” and “concurrent administration”—is still needed in the future to confirm the optimal timing of FGFR inhibitors with chemotherapy [348].

Regarding dose optimization, FIGHT-101 established and recommended pemigatinib at 13.5 mg QD (intermittent dosing, 2 weeks on/1 week off) as the dose for subsequent studies. FIGHT-202 was conducted under this regimen and validated clinical activity in cholangiocarcinoma with *FGFR2* fusions/rearrangements. In FIGHT-202, the ORR in patients with *FGFR2* fusions/rearrangements was approximately 37.0%, supporting the clinical significance of dose and patient selection [4,349]. However, clinical evidence regarding combination of pemigatinib with standard chemotherapy (such as gemcitabine/cisplatin) remains limited, currently consisting primarily of preclinical studies, early-phase/small-sample clinical reports, and reviews, with a lack of large-scale randomized controlled trials to definitively establish optimal combination doses and administration schedules [350]. Additionally, chemotherapy dose intensity (clinically used GEM doses typically around 1000 mg/m^2^, with variations of 800–1250 mg/m^2^ in different regimens) is closely associated with toxicity risk, requiring careful balance between efficacy and safety in clinical practice [351,352,353].

#### 6.1.2. Molecular Mechanisms of Radiosensitization and Clinical Translation Prospects

The radiosensitization effect of FGFR inhibitors in bladder cancer may depend on their intervention in DNA damage response and cell cycle regulation. FGFR/FGF signaling can affect multiple cell cycle checkpoints, commonly at the G1/S phase, and in specific contexts can also act on G2/M, thereby exhibiting context-dependent radiosensitivity [354,355]. Preclinical studies have shown that FGFR inhibitors can impair homologous recombination repair capacity, for instance, by prolonging the duration of γH2AX signaling and reducing RAD51 expression, thereby exacerbating radiation-induced DNA double-strand break damage and enhancing apoptosis [356,357].

Regarding clinical evidence, large-scale randomized trials confirming FGFR inhibitors as universal radiosensitizers are currently lacking. The phase III THOR study (cohort 2) compared the efficacy of erdafitinib versus pembrolizumab, with results showing no significant difference in OS (10.9 vs. 11.1 months, *p* = 0.18); PFS was numerically superior (4.4 vs. 2.7 months) but did not reach statistical significance; however, ORR was significantly improved (40.0% vs. 21.6%) [358]. Nevertheless, this study did not include radiotherapy-related controls or subgroup analyses, and thus is insufficient to validate the radiosensitization effect of FGFR inhibitors. Recent systematic reviews have also emphasized that current evidence is inadequate to support widespread combination of FGFR inhibitors with radiotherapy, and future studies will need to rely on prospective randomized trials and systematic meta-analyses to clarify their clinical value [58].

#### 6.1.3. Translational Prospects of FGFR2 Fusions in Postoperative Risk Stratification and Individualized Treatment of Cholangiocarcinoma

*FGFR2* fusions have gradually demonstrated potential value in postoperative prognostic assessment and surgical decision-making. A retrospective single-center study revealed that *FGFR2* fusion-positive patients exhibited significantly superior long-term survival compared to fusion-negative patients following curative resection, and *FGFR2* fusion was identified as an independent protective factor in multivariate analysis, suggesting its utility as a molecular biomarker for preoperative or intraoperative risk stratification [359]. Furthermore, related cohort studies have found that patients with *FGFR2* fusions or rearrangements often exhibit reprogramming of immune-related pathways and differential immune cell infiltration patterns, which are closely associated with improved clinical outcomes, further reinforcing their potential application value in individualized treatment strategies [140].

At the mechanistic level, *FGFR2* fusions can constitutively activate the RAS/MAPK and PI3K/AKT signaling pathways, promoting tumor cell proliferation and invasion. Targeted intervention of these signaling axes may alter tumor biological behavior, providing a theoretical rationale for perioperative treatment [360]. Clinical studies have demonstrated that FGFR inhibitors can partially reverse the associated phenotypes. Notably, the irreversible inhibitor futibatinib has exhibited significant efficacy in patients with *FGFR2* fusion-positive iCCA who are resistant to ATP-competitive inhibitors, suggesting that perioperative treatment regimens should be individualized and optimized based on resistance mutation profiles [191,361].

However, existing evidence predominantly originates from retrospective or small-sample studies, and large-scale prospective validation remains lacking. Molecular detection methodology also represents a critical limiting factor for clinical translation. Different detection platforms exhibit variations in coverage, sensitivity, and fusion partner identification capability: studies have shown that DNA-only NGS has a higher false-negative rate compared to RNA sequencing or FISH, and concordance across different study series is limited. Therefore, clinical practice should adopt multimodal detection or repeat validation strategies to enhance accuracy [192,362]. Overall, *FGFR2* fusions demonstrate significant value in surgical prognostic assessment and individualized treatment, but their clinical application still requires further confirmation through multicenter large-scale studies.

#### 6.1.4. Mechanisms of Action and Translational Evidence for FGFR Inhibitors Combined with Chemotherapy

Studies have demonstrated that *FGFR1* amplification or overexpression is closely associated with the EMT program and invasive/metastatic phenotypes. In multiple lung cancer models, inhibition of FGFR signaling can suppress or reverse EMT features and partially restore sensitivity to certain antitumor agents [363,364]. In the *KRAS*-mutant context, FGFR1 can serve as a compensatory pathway following inhibition of downstream KRAS signaling; therefore, the combination of FGFR inhibitors with chemotherapy or downstream pathway inhibitors has demonstrated synergistic effects in preclinical models under this context [365,366].

Another potential synergistic mechanism involves c-Myc regulation: multiple models have shown that FGFR inhibitors can downregulate c-Myc and promote tumor cell apoptosis. Taxanes can also reduce c-Myc expression in certain systems, suggesting that the two have biological synergistic potential in regulating c-Myc and related proliferation/apoptosis networks. A recent review on the association between taxanes and c-Myc has summarized supporting evidence from different models, indicating that this synergistic hypothesis warrants further validation in in vivo and clinical trials [367,368].

#### 6.1.5. Toxicity and Management Strategies for FGFR Inhibitor Combination Therapy

Multimodal combinations (FGFR inhibitor + chemotherapy ± radiotherapy) are frequently associated with on-target toxicities, including hyperphosphatemia, ocular toxicity, and fatigue. Hyperphosphatemia is the most common, with management measures including low-phosphate diet, phosphate binders, and dose adjustment or treatment discontinuation when necessary. Ocular toxicity (such as corneal epithelial changes and retinopathy) requires regular monitoring before and during treatment to enable early detection and intervention [14,369].

Clinical experience and guidelines indicate that combination therapy has a higher incidence of toxicity compared to monotherapy, but most adverse events can be controlled through supportive care and dose adjustment. Current toxicity management is largely experience-based and lacks large-scale prospective validation. Multiple reviews have emphasized that establishing standardized management frameworks (early monitoring, multidisciplinary collaboration, and timely intervention) can significantly reduce treatment discontinuation rates and improve adherence, though specific outcomes vary across studies [370,371]. Therefore, toxicity management of combination therapy requires multidisciplinary collaboration to ensure a balance between efficacy and safety and to improve patient outcomes.

### 6.2. Synergistic Mechanisms of FGFR-Targeted Therapy and Immunotherapy

The combination of FGFR-targeted therapy and immunotherapy is considered a cutting-edge strategy that can overcome the limitations of monotherapy by dually modulating the tumor microenvironment and enhancing antitumor immune responses. Current research focuses include: the mechanistic role of FGFR signaling in immune regulation, evidence from preclinical combination studies, exploration of adoptive cell therapy and cancer vaccines, and synergistic effects of gene regulation in specific cancer types. Based on multi-omics data and preclinical models, these strategies hold promise for improving therapeutic efficacy and reversing resistance.

#### 6.2.1. Regulatory Mechanisms of FGFR Signaling on the Immune Microenvironment

FGFR signaling plays a pivotal regulatory role in shaping the immune microenvironment in bladder cancer. *FGFR3* activation or mutation is particularly common in bladder cancer, with an incidence of 50–70% in non-muscle-invasive cases and approximately 10–15% in muscle-invasive tumors, with an overall reported rate of 15–20% [78,372]. Mechanistic studies have revealed that FGFR3 can remodel the immune microenvironment through regulation of PD-L1 stability and the IFN-γ signaling pathway. Jing et al. found that FGFR3 activation promotes NEDD4-mediated ubiquitination and degradation of PD-L1, leading to PD-L1 downregulation and reduced CD8^+^ T cell infiltration; whereas FGFR3 inhibition can block this process, upregulate PD-L1, and enhance anti-PD-1 therapeutic efficacy in mouse models, providing a molecular rationale for combining FGFR inhibitors with immune checkpoint inhibitors [138]. Additionally, FGFR signaling activation can suppress the IFN-γ–JAK/STAT response, limiting cytokine and chemokine expression, thereby attenuating the recruitment and function of dendritic cells and natural killer cells; whereas FGFR inhibition can restore IFN-γ pathway activity and enhance immune activation [132,136]. These findings reveal the bidirectional role of FGFR signaling in immune regulation, although most remain based on in vitro and animal studies and await clinical validation.

Retrospective analyses have indicated that *FGFR3* alterations are typically associated with a “non-T cell-inflamed” tumor phenotype, characterized by insufficient CD8^+^ T cell infiltration and poor response to ICI monotherapy. However, this association exhibits significant heterogeneity: in low-grade or well-differentiated lesions, *FGFR3* mutations are sometimes associated with better prognosis, whereas in high-grade tumors or those harboring co-mutations such as *TP53* and *PIK3CA*, they are frequently associated with immune evasion and poor ICI response [108,372,373]. These findings underscore that the immune regulatory effects of FGFR3 are jointly influenced by pathological grade, molecular context, and immune ecology.

In summary, FGFR3 influences tumor immune phenotype through regulation of PD-L1 stability and the IFN-γ signaling pathway, serving as an important molecular hub determining the immune microenvironment in bladder cancer. Its biological characteristics provide a theoretical foundation for FGFR status-based combination immunotherapy, but further validation in prospective clinical studies with clearly defined molecular stratification is still needed to clarify its translational potential and optimal combination strategies.

#### 6.2.2. FGFR Inhibition-Mediated Immune Remodeling and Its Translational Potential in Combination Immunotherapy

Extensive preclinical studies have demonstrated that FGF/FGFR pathway inhibition can remodel the tumor immune microenvironment, thereby enhancing the antitumor efficacy of ICIs. Mechanistically, FGFR blockade can restore or enhance IFN-γ signaling pathway activity, upregulate antitumor immunity-related genes, and promote antigen presentation; additionally, it can induce vascular normalization, reduce immunosuppressive macrophage infiltration, and reactivate the IFN-γ pathway, thereby generating synergistic effects with anti-PD-1 or anti-PD-L1 therapy [136,336]. It should be noted that some agents used in combination immunotherapy (such as lenvatinib) are multi-target inhibitors, and their immunomodulatory effects are not solely mediated by FGFR inhibition. Recent clinical and real-world studies both suggest that careful distinction of the contributions of different targets is required when interpreting the efficacy of lenvatinib-ICI combinations [337,374].

Multiple animal model studies have further validated this synergistic effect: FGFR inhibitors combined with PD-1/PD-L1 inhibitors can significantly increase CD8^+^ T cell infiltration, reduce immunosuppressive cell populations, and promote TCR clonal expansion, thereby achieving superior antitumor activity and survival benefit compared to monotherapy. For example, in *FGFR2*-driven lung cancer and *FGFR3*-driven urothelial carcinoma models, erdafitinib or other FGFR inhibitors combined with anti-PD-1 all demonstrated significant synergistic effects [132,336]. However, efficacy varied significantly across different models, and some mouse studies also observed increased combination therapy-related toxicity, suggesting that clinical translation requires thorough evaluation of safety and tolerability.

Clinically, the FIGHT-207 study demonstrated that pemigatinib achieved a monotherapy ORR of 26.5% in select solid tumors with *FGFR* alterations, suggesting that FGFR inhibition has definitive antitumor activity. However, clinical data on combinations with ICIs remain limited at present. Future studies should systematically evaluate the synergistic potential of FGFR inhibition and immunotherapy through stratified early- to mid-phase clinical trials, and validate their long-term impact on PFS, OS, and safety in randomized phase III trials [224,336].

#### 6.2.3. Exploratory Combination of Cellular Therapies Such as CAR-T and TCR-T with FGFR-Targeted Therapy

In cholangiocarcinoma, *FGFR2* fusions not only provide a target for small molecule inhibitors but also lay the foundation for neoantigen-directed cell therapies such as TCR-T. Research by White et al. demonstrated that TILs isolated from patients with metastatic cholangiocarcinoma can specifically recognize *FGFR2* fusion peptides, directly demonstrating that these sites possess natural immune accessibility and providing critical evidence for TCR-T design [375]. Currently, FGFR-targeted cell therapy research remains largely in preclinical or early clinical phases. Previous studies have indicated that FGFR-targeted CAR-T can be developed and have demonstrated potential in some solid tumor models; however, due to differences in tissue distribution and expression profiles among different subtypes of the FGFR family, *FGFR2* fusion-directed CAR-T development faces greater challenges [338,376,377].

Regarding combination strategies, theoretical and preliminary studies suggest that FGFR-TKIs can enhance CAR-T/TCR-T through the following pathways: first, by weakening tumor-dependent pathways and exposing more antigens; second, by remodeling the tumor microenvironment (reducing MDSCs/TAMs and improving vascular permeability), thereby promoting exogenous T cell infiltration and persistence; and third, by achieving precise elimination through fusion site-specific TCRs. Although these mechanisms are supported in vitro and in limited animal models, systematic preclinical data on pemigatinib or other selective FGFR-TKIs combined with CAR-T/TCR-T remain insufficient, and existing efficacy evidence should be cautiously interpreted [375,378].

Although this strategy possesses potential advantages through targeting of specific fusion proteins and improvement in the immune microenvironment, its limitations are also significant: first, fusion partner diversity leads to antigenic heterogeneity; second, cell therapy in solid tumors still faces challenges of insufficient infiltration, T cell exhaustion, and immunosuppressive barriers; third, safety concerns may compound the metabolic and electrolyte toxicities of FGFR-TKIs. Future efforts require establishment of a systematic preclinical validation framework and comprehensive evaluation of safety and durable efficacy through phased clinical trials progressing from small to expanded sample sizes [338,379].

#### 6.2.4. Application Prospects of Tumor Vaccines in FGFR-Positive Tumors

Tumor vaccines can provide novel strategies for the treatment of FGFR-positive tumors by inducing specific immune responses. Animal model studies have demonstrated that vaccines based on xenogeneic FGFR1 (pxFR1) can significantly inhibit tumor growth and exert antitumor effects through activation of CD8^+^ T cells and antibody responses [380]. Furthermore, FGFR1 as an immune target can be combined with FGFR inhibitors to enhance T cell responses and memory effects [381]. These findings suggest that FGFR-related antigens possess certain immunogenicity, providing a theoretical basis for the development of FGFR-targeted tumor vaccines.

However, compared to vaccine research for molecular targets such as HER2, EGFR, and NY-ESO-1, FGFR vaccine development is notably lagging. HER2 vaccines have entered multiple phase I/II clinical trials in breast cancer and demonstrated certain immunological activity; EGFR vaccines have also been explored in non-small-cell lung cancer, and although efficacy remains limited, translational experience has been accumulated; while cancer testis antigen vaccines such as NY-ESO-1 and MUC1 have demonstrated durable T cell responses in various solid tumors [382,383,384]. In contrast, FGFR vaccines still lack systematic preclinical and clinical validation, with existing evidence remaining primarily at the level of animal experiments.

Overall, vaccine development for FGFR-positive tumors remains in its nascent stages, with significant research gaps. Its potential value is more likely to be reflected in adjuvant therapy or recurrence prevention rather than monotherapy, and requires interdisciplinary collaboration and multicenter clinical trials to further advance the field.

## 7. Conclusions

Although extensive research has revealed the biological diversity of FGFR aberrations in various solid tumors and their feasibility as potential drug targets, there remains a lack of systematic, pan-cancer evidence integration that comprehensively compares different FGFR aberration types, resistance profiles, and optimal clinical management pathways. To address this gap, this review systematically reviews and synthesizes key evidence from recent years regarding the molecular mechanisms, clinical development progress, and translational challenges of FGFR aberration-driven solid tumors. Multi-omics analyses and functional studies consistently demonstrate that in several cancer types, FGFR aberrations can serve as definitive oncogenic driver events and significantly influence the heterogeneity in pharmacological sensitivity and clinical benefit to FGFR-targeted strategies. With the successive approval of targeted drugs such as pemigatinib, erdafitinib, and futibatinib and their demonstration of considerable efficacy in clinical trials, FGFR-targeted therapy has transitioned from concept to partial clinical benefit. However, the heterogeneity in real-world efficacy, complex resistance mechanisms, and lack of biomarkers indicate that substantial work remains in the clinical translation of this field.

First, different types of FGFR aberrations exhibit significant differences in signal dependency and drug sensitivity. Fusions and rearrangements typically lead to ligand-independent activation, whereas amplifications or point mutations display varying degrees of dependency. Therefore, precise stratification based on aberration type, fusion partners, and co-mutation profiles is critical for achieving personalized therapy. Although qualitative insights into pathway hyperactivation are presented here, a quantitative meta-analysis of signaling intensity remains unfeasible due to the lack of standardized data. Existing evidence relies largely on disparate assays (e.g., Western blot) with varying baseline conditions, preventing a unified comparison of MAPK/ERK or PI3K/AKT activation across different FGFR genotypes. Future efforts must therefore prioritize standardized phosphoproteomic quantification to bridge the gap between mechanistic characterization and clinical stratification.

Second, acquired resistance is widespread. Secondary kinase domain mutations, compensatory activation of downstream pathways or other RTKs represent the major mechanisms. Irreversible inhibitors and multi-target combination strategies developed specifically for these mechanisms have shown partial potential to overcome resistance, but polyclonal resistance and heterogeneity remain to be addressed. Liquid biopsy-assisted dynamic monitoring and sequential therapy will become future directions.

Additionally, FGFR inhibition exerts dual regulatory effects on the tumor immune microenvironment. It can enhance antitumor immune responses but may also upregulate PD-L1 expression in certain contexts. Precision combination designs based on molecular and immune features are expected to improve response rates to immune checkpoint inhibitors, while FGF-VEGF signaling crosstalk also suggests potential value in combining with antiangiogenic therapy.

Future research should focus on three aspects: ① establishing a standardized biomarker system that integrates tissue, ctDNA, and multi-omics data; ② optimizing combination and sequential strategies to delay resistance development; ③ promoting the development of new inhibitors with high selectivity or covalent binding properties. The introduction of artificial intelligence and multimodal data analysis is expected to further enhance the predictive and monitoring precision of FGFR-targeted therapy.

Overall, research on *FGFR* aberration-driven tumors is transitioning from mechanistic exploration toward clinical translation. Through molecular stratification, dynamic monitoring, and combination innovation, FGFR-targeted therapy is expected to achieve reproducible clinical benefit in a broader range of cancer types.

## Figures and Tables

**Figure 1 cancers-18-00089-f001:**
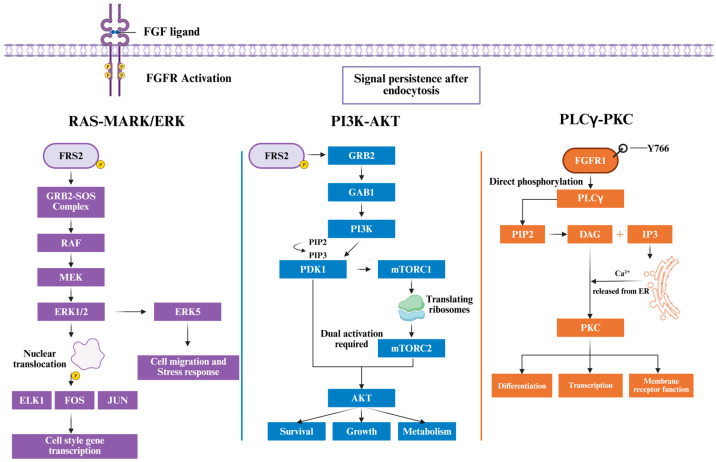
Major Signaling Cascades Downstream of FGFR Activation. A schematic representation of three major signaling cascades activated by FGFR: the RAS-MAPK/ERK pathway (purple) regulating transcription and mediating cell migration and stress responses, the PI3K-AKT pathway (blue) controlling cell survival, growth, and metabolism, and the PLCγ-PKC pathway (orange) mediating differentiation, transcription, and membrane receptor function. Detailed pathway descriptions and abbreviations are provided in the main text.

**Figure 2 cancers-18-00089-f002:**
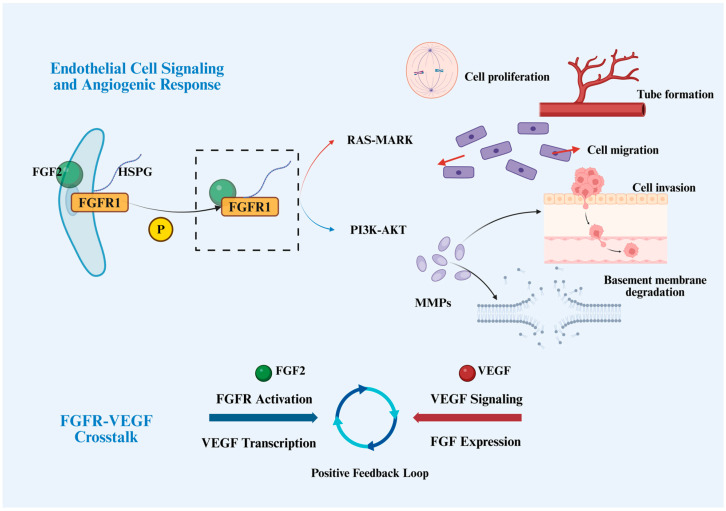
FGFR Signaling Regulates Tumor Angiogenesis Through Endothelial Cell Activation and VEGF Crosstalk. Upper panel: Tumor-derived FGF2 (green sphere) binds to FGFR1 (orange) and HSPG co-receptor (blue glycan chains) on endothelial cells, inducing receptor dimerization and autophosphorylation (marked as P). Activated FGFR1 triggers RAS-MAPK-ERK (red) and PI3K-AKT (blue) pathways, coordinately driving pro-angiogenic effects. Lower panel: FGFR-VEGF positive feedback loop amplifies angiogenic signaling. FGF2 (left, green sphere) activates FGFR to upregulate VEGF transcription and secretion. Secreted VEGF (right, red sphere) activates VEGFR, reciprocally enhancing FGF expression.

**Figure 3 cancers-18-00089-f003:**
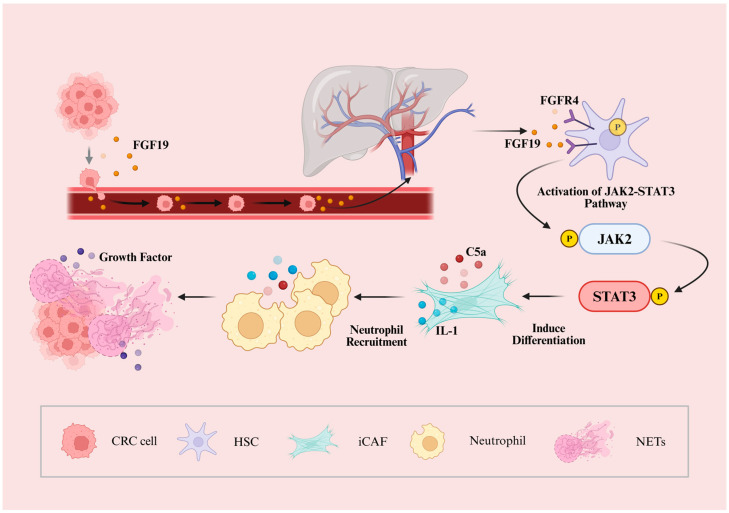
FGF19-FGFR4 axis promotes colorectal cancer liver metastasis through neutrophil extracellular trap formation. Tumor-derived FGF19 (orange sphere) activates FGFR4 on hepatic stellate cells (HSCs), triggering JAK2-STAT3 signaling and inducing HSC differentiation into inflammatory cancer-associated fibroblasts (iCAFs). iCAFs secrete IL-1 (blue sphere), complement C5a (red sphere), and chemokines to recruit neutrophils and promote NET formation. NETs facilitate colorectal cancer liver metastasis by capturing circulating tumor cells (CTCs) and releasing growth-promoting factors (purple sphere), thereby enhancing tumor cell adhesion, survival, and proliferation in the liver. CRC, colorectal cancer; HSC, hepatic stellate cell; iCAF, inflammatory cancer-associated fibroblast; NETs, neutrophil extracellular traps; CTCs, circulating tumor cells.

**Figure 4 cancers-18-00089-f004:**
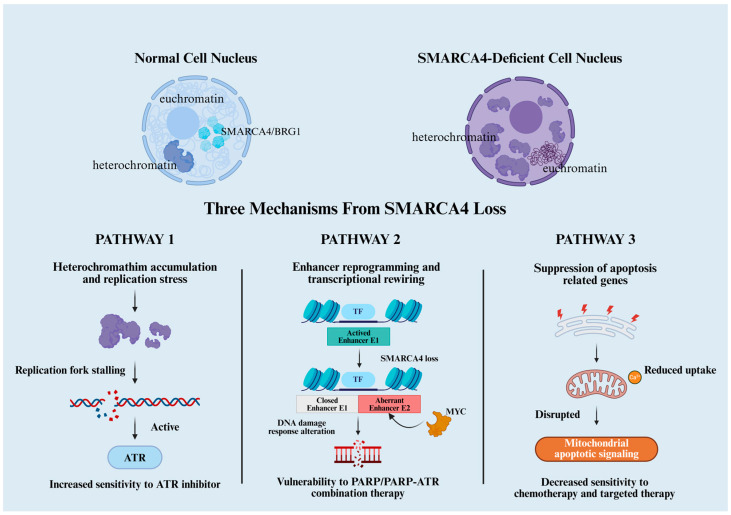
Molecular Mechanisms Underlying SMARCA4/BRG1 Loss-Mediated Therapeutic Response in Cancer. Upper panel: Chromatin organization in normal versus SMARCA4-deficient cell nuclei. The SMARCA4/BRG1 complex maintains euchromatin-heterochromatin balance in normal cells (**left**). SMARCA4 loss (**right**) causes global heterochromatin accumulation and reduced chromatin accessibility. Pathway 1 (**left**): Heterochromatin accumulation and replication stress. SMARCA4 deficiency promotes excessive heterochromatin formation (purple clusters), causing replication fork stalling and DNA damage. Persistent replication stress activates ATR signaling, creating oncogenic addiction to ATR-mediated DNA damage response and conferring sensitivity to ATR inhibitors. Pathway 2 (**center**): Enhancer reprogramming and transcriptional rewiring. SMARCA4 loss induces enhancer reprogramming, converting active enhancers to closed or aberrantly activated states. This activates MYC signaling and alters DNA damage response (DDR) gene expression, conferring vulnerability to PARP inhibitors. Pathway 3 (**right**): Suppression of apoptosis-related genes. SMARCA4 deficiency downregulates pro-apoptotic genes, disrupting ER-to-mitochondria Ca^2+^ transfer. Impaired mitochondrial Ca^2+^ uptake compromises apoptotic signaling, reducing sensitivity to chemotherapy and FGFR-targeted therapy.

**Table 1 cancers-18-00089-t001:** FGFR Alterations Across Major Solid Tumor Types: Frequency, Predominant Alteration Type, and Molecular Characteristics.

Cancer Type	FGFR Alteration Frequency (%)	Predominant FGFR Alteration Type	Common Subtypes/Mutation Sites or Fusion Partners
Intrahepatic cholangiocarcinoma	10–16	FGFR2 fusion	BICC1, AHCYL1, PPHLN1
Bladder cancer/Urothelial carcinoma	15–20	FGFR3 mutation	S249C, R248C, Y373C
Breast cancer	14–18	FGFR1 amplification	ER(+)
Endometrial cancer	10–12	FGFR2 mutation	S252W, N549K
Lung squamous cell carcinoma	15–20	FGFR1 amplification	NA
Glioblastoma	4–8	FGFR3 fusion	TACC3

Abbreviations: FGFR, fibroblast growth factor receptor; ER, estrogen receptor; NA, not applicable.

**Table 2 cancers-18-00089-t002:** Clinical Efficacy and Trial Design of FGFR Inhibitors Across Different Tumor Types and Molecular Alterations.

Drug	Cancer Type	Trial Name	Study Design	Key Efficacy Outcomes	References
Pemigatinib	Cholangiocarcinoma with FGFR2 fusion/rearrangement	FIGHT-202	Multicenter, open-label, phase II	ORR 37.0%; mPFS 7.0 months; mOS 17.5 months	[4,258]
	Solid tumors with FGFR alterations (multiple cancer types)	FIGHT-207	Phase II basket trial with cohorts stratified by molecular alteration type	ORR 26.5% in FGFR fusion patients; only 9.4% in non-kinase domain mutations	[224]
	Advanced cholangiocarcinoma with FGFR2 rearrangement	FIGHT-302 (ongoing)	Multicenter, open-label, randomized, phase III	Primary endpoint: PFS; secondary endpoints: OS, ORR, safety, etc.	[259]
Erdafitinib	Urothelial carcinoma with FGFR alterations	BLC2001	Single-arm, multicenter, phase II	ORR 40%; mDOR 5.6 months; mPFS 5.5 months	[5,260]
	Urothelial carcinoma with FGFR alterations	THOR	Multicenter, open-label, randomized controlled, phase III	mOS 12.1 months vs. 7.8 months (*p* = 0.005)	[261]
	Solid tumors with FGFR alterations (multiple cancer types)	RAGNAR	Multicenter, single-arm basket, phase II	ORR 29% in overall population; mDOR 6.9 months	[262]
Futibatinib	Cholangiocarcinoma with FGFR2 fusion/rearrangement	FOENIX-CCA2	Multicenter, single-arm, phase II	ORR 42%; mPFS 9.0 months; mOS 21.7 months	[191]
Lirafugratinib (RLY-4008)	Solid tumors with FGFR2 alterations	ReFocus (ongoing)	Multicenter, open-label, phase I/II	Primary endpoints: safety/tolerability, ORR; secondary endpoints: PFS, OS, DCR, DOR	[263,264]
Bemarituzumab	FGFR2b-positive gastric cancer/GEJ cancer	FIGHT	Multicenter, randomized, double-blind, phase II	mPFS 9.5 months vs. 7.4 months (*p* = 0.07)	[226]
Vofatamab (B-701)	Metastatic urothelial carcinoma	FIERCE-22	Multicenter, open-label, phase Ib/II	ORR 32%; mPFS 4.7 months	[227]
Aprutumab ixadotin	Solid tumors with high FGFR2 expression	NCT02368951	Open-label, multicenter, phase I	Terminated due to severe proteinuria/hematologic toxicity	[229]

Abbreviations: ORR, objective response rate; mPFS, median progression-free survival; mOS, median overall survival; mDOR, median duration of response; DCR, disease control rate; DOR, duration of response; GEJ, gastroesophageal junction.

## Data Availability

No new data were generated or analyzed in this study. All data discussed in this review are derived from previously published studies and publicly available databases cited within the manuscript.

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
