# Peer review of "FGFR Aberrations in Solid Tumors: Mechanistic Insights and Clinical Translation of Targeted Therapies"

_cancers, 2025, doi:10.3390/cancers18010089_

Round 1

Reviewer 1 Report

Comments and Suggestions for Authors

The review manuscript is an extensive and detailed description of the mechanisms and roles of FGFRs in malignancies, and the ongoing achievements of the detection and therapeutic applications to control FGFR-driven tumorigenesis and progression. The basic description is textbook-like and lacks mention less well-known topics such as FGFRL1 (FGFR5).

The aim is ambitious and leads to heavy reading of the text. However, the text is analytic in places instead of only being a collection and listing of the details of the related molecular mechanisms of FGFRs and their exploitation in the development of therapeutic application. The description of the current status of development and clinical testing of new compounds against cancers driven by FGFR aberrations is detailed and useful summary of the situation. Also, the discussion on the current diagnostic tools and side effects of therapies, and future possibilities is useful.

The text includes some printing errors which might need manual detection and correction.

Author Response

Comment1: The basic description is textbook-like and lacks mention less well-known topics such as FGFRL1 (FGFR5).

Response1: 

We thank the reviewer for this valuable suggestion. In response, we have substantially expanded the manuscript to incorporate the atypical FGFR family member FGFRL1 (FGFR5), thereby extending the review beyond the classical kinase-active receptors FGFR1–4.

Specifically, we added a description of FGFRL1 structure and function as a decoy/pseudo-receptor lacking a tyrosine kinase domain (Lines 104–112). We further included genetic and experimental evidence highlighting its essential role in organogenesis, particularly kidney and diaphragm development (Lines 156–167), as well as a mechanistic discussion of FGFRL1 as a molecular rheostat modulating FGF–FGFR signaling amplitude and spatial range (Lines 184–192).

Finally, we introduced a new subsection summarizing emerging evidence for FGFRL1 involvement in solid tumors, including associations with tumor progression, patient outcome, and therapy resistance in multiple cancer types (Lines 242–252). These additions directly address the reviewer’s concern and provide a more comprehensive overview of the FGF–FGFR signaling network.

Reviewer 2 Report

Comments and Suggestions for Authors

Major revision recommendation

The manuscript presents a comprehensive overview of FGFR aberrations; however, major revision is required to strengthen mechanistic and translational clarity. The authors should provide deeper integration of pathway interactions, particularly FGFR–RTK crosstalk, expand discussion of resistance-associated mutations, and include more quantitative comparisons of aberration frequency across tumor types. Additionally, clearer linkage between molecular mechanisms and therapeutic implications—including inhibitors, ADCs, and nanodelivery approaches—is needed to enhance coherence and interpretability.

  1. The manuscript outlines the structural domains of FGFR1–4; however, can the authors include comparative structural modeling or additional citations clarifying how specific extracellular-domain mutations disrupt autoinhibition?
  2. The description of ligand–receptor specificity would benefit from quantifying binding affinities or including structural data showing how Klotho proteins alter FGFR selectivity. Could the authors expand this section?
  3. The pan-cancer prevalence values for FGFR aberrations are summarized, but could the authors provide a more granular breakdown by ethnicity or sequencing platform sensitivity?
  4. The manuscript discusses pathway hyperactivation but remains descriptive. Could the authors integrate quantitative data on fold-changes in MAPK/AKT signaling driven by specific aberrations?
  5. For FGFR2 fusions, additional mechanistic insight into fusion-protein trafficking or oligomerization would strengthen the discussion.
  6. The authors state that FGFR signaling suppresses the IFN-γ → STAT1/IRF1 axis; could they provide tumor-type-specific examples or supporting datasets?
  7. The section on CAF heterogeneity references single-cell evidence; can spatial-transcriptomic validation be added for specific tumor types?
  8. The review describes the role of FGFR in metastatic niche formation. Could the authors compare FGFR-driven niches with those induced by MET or CXCR4 signaling?
  9. The manuscript mentions NET-mediated metastasis downstream of FGFR4; can additional mechanistic steps or experimental models be cited to strengthen this claim?
  10. For nanomedicine platforms, could the authors discuss the translational gap related to EPR variability among human tumors compared to preclinical models?

Author Response

Comment1: The manuscript outlines the structural domains of FGFR1–4; however, can the authors include comparative structural modeling or additional citations clarifying how specific extracellular-domain mutations disrupt autoinhibition?

Response1: We thank the reviewer for this constructive suggestion. In response, we expanded the manuscript to include additional structural and biochemical evidence illustrating how extracellular-domain mutations disrupt FGFR autoinhibition.

Specifically, we added a detailed discussion of recurrent FGFR3 extracellular hotspot mutations (R248C, S249C, and G370C) in urothelial carcinoma, which cluster within the Ig-like domains or the IgII–IgIII linker region. Drawing on biochemical analyses of the corresponding germline mutations causing thanatophoric dysplasia type I, we explain how cysteine substitutions introduce unpaired cysteines that promote aberrant intermolecular disulfide bond formation and ligand-independent covalent receptor dimerization.

In addition, we integrated crystallographic and biophysical studies describing the autoinhibitory role of the D1–acid box module and the requirement for HSPG-mediated FGF–FGFR assembly. Together, these data support a mechanistic model in which Cys-substituting FGFR3 mutations stabilize covalent homodimers that partially bypass D1–acid box–mediated autoinhibition and the normal ligand-dependent dimerization process, resulting in constitutive kinase activation (Lines 668–682). These additions directly address the reviewer’s request for greater structural and mechanistic clarity.

Comment2: The description of ligand–receptor specificity would benefit from quantifying binding affinities or including structural data showing how Klotho proteins alter FGFR selectivity. Could the authors expand this section?

Response2: We thank the reviewer for this helpful suggestion. In response, we expanded the section on ligand–receptor specificity to incorporate structural and functional evidence illustrating how Klotho proteins modulate FGFR selectivity and binding affinity.

Specifically, we added a discussion of Klotho proteins as essential co-receptors that reshape FGFR ligand specificity, with emphasis on the FGF19–FGFR4–βKlotho complex in hepatic tissues (Lines 124–135). We now cite structural and biochemical studies demonstrating that βKlotho stabilizes receptor conformations that markedly enhance FGF19 binding affinity to FGFR4 compared with the absence of βKlotho, thereby conferring receptor and tissue specificity. In contrast, we clarify that canonical FGFs such as FGF1 and FGF2 bind FGFRs independently of Klotho proteins and exhibit broader receptor specificity.

In addition, we expanded the mechanistic description of how ligand engagement of the extracellular Ig-like domains promotes receptor dimerization, which is further stabilized by Klotho proteins, leading to activation of downstream signaling pathways including RAS/MAPK and PI3K/AKT (Lines 141–147). These revisions address the reviewer’s request by providing greater structural and affinity-based context for Klotho-dependent FGFR selectivity.

Comment3: The pan-cancer prevalence values for FGFR aberrations are summarized, but could the authors provide a more granular breakdown by ethnicity or sequencing platform sensitivity?

Response3: We thank the reviewer for this important point. In response, we expanded the manuscript to clarify how both sequencing platform sensitivity and patient ethnicity influence reported pan-cancer prevalence estimates of FGFR aberrations.

First, we added a dedicated discussion addressing assay-dependent detection bias (Lines 210–217). We now highlight that DNA-based hybrid-capture panels and whole-exome sequencing frequently under-detect FGFR rearrangements compared with RNA-based NGS or anchored multiplex PCR, which show higher sensitivity for in-frame and rare-partner fusions. Conversely, high-depth targeted NGS and array-CGH exhibit greater sensitivity for low-level FGFR copy-number gains, whereas cfDNA-based NGS may miss subclonal or low–allele-fraction FGFR events due to tumor shedding dynamics.

Second, we incorporated an ethnicity- and geography-informed comparison of available pan-cancer datasets (Lines 289–296). We note that overall FGFR alteration prevalence is broadly comparable between Chinese and predominantly European/North American cohorts, while the distribution of FGFR-altered tumor types differs by population, with Chinese datasets enriched for gastric and colorectal cancers and Western cohorts dominated by urothelial, breast, and endometrial tumors. We also emphasize the current scarcity of robust prevalence data from African, South Asian, and Latin American populations.

Together, these additions provide a more granular and contextualized interpretation of pan-cancer FGFR prevalence estimates, directly addressing the reviewer’s concern.

Comment4: The manuscript discusses pathway hyperactivation but remains descriptive. Could the authors integrate quantitative data on fold-changes in MAPK/AKT signaling driven by specific aberrations?

Response4: We thank the reviewer for this thoughtful comment. While comprehensive fold-change quantification across FGFR aberrations remains challenging, we have revised the manuscript to incorporate available quantitative context and to transparently outline current limitations.

First, we added a new paragraph summarizing existing phosphoproteomic and immunoblot evidence showing that FGFR amplifications, activating mutations, and oncogenic fusions consistently activate MAPK/ERK and PI3K/AKT signaling relative to FGFR–wild-type controls (Lines 466–472). Importantly, we note that quantitative analyses across different FGFR isoforms, fusion partners, and cellular contexts reveal substantial variability in signaling magnitude, indicating that no single fold-change value can be generalized across aberration classes or tumor types.

Second, we expanded the Conclusion to explicitly address the methodological constraints that currently preclude a formal quantitative meta-analysis (Lines 1716–1723). We explain that heterogeneous assay platforms, non-standardized baselines, and variable normalization strategies prevent reliable cross-study comparison of MAPK/ERK and PI3K/AKT activation intensity.

Together, these revisions integrate the available quantitative evidence while clearly defining the boundaries of current knowledge, and highlight the need for standardized phosphoproteomic approaches to enable robust, clinically relevant quantification in future studies.

Comment5: For FGFR2 fusions, additional mechanistic insight into fusion-protein trafficking or oligomerization would strengthen the discussion.

Response5: We thank the reviewer for this valuable suggestion. In response, we expanded the discussion of FGFR2 fusion oncogenic mechanisms to include emerging insights into fusion-driven oligomerization and intracellular trafficking(Lines 705–722).

Specifically, we added a new section describing how FGFR2 fusion partners encode multimerization modules—such as coiled-coil or Sterile Alpha Motif domains—that promote ligand-independent dimerization or higher-order clustering, thereby stabilizing constitutive kinase activation. We further discuss representative fusion partners, including TACC3 and BICC1, highlighting how their structural domains not only drive aberrant oligomerization but also redirect fusion proteins to distinct subcellular compartments, such as centrosomal, mitotic spindle, or endosomal locations.

Importantly, we now emphasize that altered trafficking dynamics—affecting receptor recycling, degradation, and spatial signaling—can prolong FGFR kinase activity and reshape downstream signaling amplitude and duration. Together, these additions provide mechanistic depth beyond kinase activation alone and highlight fusion-driven oligomerization and trafficking as potential therapeutic vulnerabilities complementary to kinase inhibition.

Comment6: The authors state that FGFR signaling suppresses the IFN-γ → STAT1/IRF1 axis; could they provide tumor-type-specific examples or supporting datasets?

Response6: We thank the reviewer for this insightful comment. In response, we expanded the manuscript to include tumor‑type‑specific datasets supporting suppression of the IFN‑γ–STAT1/IRF1 axis by FGFR signaling(Lines 533–544).

Specifically, we added evidence from renal cell carcinoma and multiple solid‑tumor models demonstrating that constitutive FGFR signaling attenuates IFN‑γ–induced STAT1 phosphorylation and downstream interferon‑stimulated genes, including IRF1, CXCL10, B2M, and PD‑L1, while pharmacologic or genetic FGFR inhibition restores this pathway. We further cite studies showing that this inhibitory effect of FGF/FGFR activation on interferon signaling is observed across diverse cellular systems, supporting a conserved biological mechanism rather than a model‑restricted phenomenon.

In addition, we incorporated bladder cancer–specific examples illustrating how FGFR3 activation intersects with IFN‑γ signaling by modulating PD‑L1 stability and turnover, including in FGFR3–TACC3 fusion contexts. By contrast, we note that data from FGFR2‑fusion intrahepatic cholangiocarcinoma remain limited and partially inconsistent, underscoring the need for tumor‑specific interpretation. These additions directly address the reviewer’s request for tumor‑resolved evidence.

Comment7: The section on CAF heterogeneity references single-cell evidence; can spatial-transcriptomic validation be added for specific tumor types?

Response7: We thank the reviewer for this important suggestion. In response, we expanded the CAF heterogeneity section to incorporate spatial‑transcriptomic validation across multiple tumor types, integrated with single‑cell RNA‑sequencing data(Lines 570–587).

Specifically, we added evidence from pancreatic ductal adenocarcinoma demonstrating that inflammatory CAF (iCAF), myofibroblastic CAF (myCAF), and antigen‑presenting CAF (apCAF) programs occupy distinct spatial niches and display characteristic proximity to tumor and immune compartments, as revealed by integrated spatial and single‑cell profiling. We further incorporated examples from breast, colorectal, lung, and skin cancers, where combined spatial transcriptomics and scRNA‑seq delineate conserved CAF subclasses while uncovering tumor‑specific spatial organization and neighborhood interactions.

These additions show that CAF heterogeneity is not only transcriptionally defined but also spatially organized in a context‑dependent manner, strengthening the biological relevance of CAF subtypes across tumor microenvironments and directly addressing the reviewer’s request.

Comment8: The review describes the role of FGFR in metastatic niche formation. Could the authors compare FGFR-driven niches with those induced by MET or CXCR4 signaling?

Response8: We thank the reviewer for this insightful suggestion. In response, we expanded the manuscript to include a comparative discussion of FGFR‑driven metastatic niches versus those shaped by HGF/MET and CXCL12/CXCR4 signaling (Lines 644–659).

Specifically, we added a paragraph contrasting the classical roles of MET signaling as a driver of stromal‑mediated invasive growth and EMT, and CXCR4 signaling as a chemotactic axis guiding organ‑specific tumor cell homing, with the broader functional scope of FGFR signaling. We now highlight that, beyond promoting invasion, FGFR signaling actively remodels the metastatic microenvironment by modulating innate immune composition, endothelial activation, and angiogenic programs, thereby facilitating intravasation and metastatic seeding.

In addition, we emphasize points of functional cooperation and crosstalk, noting that FGFR activity intersects with chemokine‑driven trafficking and cooperates with MET signaling to reinforce survival pathways and adaptive resistance. Collectively, these additions clarify that FGFR‑driven niches function as integrative hubs that coordinate immune, vascular, and RTK‑dependent processes, distinguishing them from more narrowly defined chemotactic or invasive signaling axes.

Comment9: The manuscript mentions NET-mediated metastasis downstream of FGFR4; can additional mechanistic steps or experimental models be cited to strengthen this claim?

Response9: We thank the reviewer for this important request. In response, we strengthened the mechanistic framework and experimental support for FGFR4‑dependent, NET‑mediated metastasis (Lines 629–637).

Specifically, we added evidence from preclinical colorectal cancer liver metastasis models demonstrating a causal FGF19–FGFR4–iCAF–NET axis. We now cite studies showing that genetic silencing of FGF19 or pharmacologic inhibition of FGFR4 or JAK2 markedly reduces iCAF polarization and hepatic NET deposition. Moreover, disruption of neutrophils or NET formation—through neutrophil depletion, DNase I treatment, or the use of Padi4‑deficient mice—significantly suppresses metastatic outgrowth.

We further emphasize functional data indicating that NETs not only trap circulating tumor cells but also provide proliferative and metabolic cues that support micrometastatic expansion, consistent with broader evidence implicating NETs in distant seeding and reactivation of dormant cancer cells. Together, these additions provide mechanistic depth and experimental validation for FGFR4‑driven, NET‑mediated metastatic niche formation.

Comment10: For nanomedicine platforms, could the authors discuss the translational gap related to EPR variability among human tumors compared to preclinical models?

Response10: We thank the reviewer for this important translational question. In response, we expanded the nanomedicine section to explicitly address the limitations and heterogeneity of the EPR effect in human tumors compared with preclinical models (Lines 911–926).

Specifically, we added a new paragraph noting that the pronounced EPR effect commonly observed in rapidly growing murine xenografts is substantially weaker and far more heterogeneous in human solid tumors. We now cite clinical and preclinical imaging studies using radiolabeled or MRI/PET‑tracked nanomedicines demonstrating that, in patients, only a limited and highly variable fraction of the injected dose accumulates within tumor tissue, with variability shaped by tumor type, anatomical location, stromal composition, vascular normalization status, and prior therapies.

Importantly, we further discuss how this discrepancy between preclinical and clinical EPR behavior likely contributes to the modest overall survival benefits observed for many EPR‑dependent nanomedicines in late‑phase clinical trials, despite strong efficacy in mouse models. Finally, we outline emerging strategies to mitigate this translational gap, including functional EPR imaging for patient stratification, transient vascular modulation to improve perfusion, and the development of actively targeted or stimuli‑responsive nanocarriers that are less dependent on passive EPR‑based delivery.

Reviewer 3 Report

Comments and Suggestions for Authors

The review article by He et al provides a comprehensive review of FGFRs and their roles in cancer. It is very well organized and is very clearly written. Overall, it is an excellent review. It will be of use to investigators who need to understand the roles of FGFRs in cancer and the status of efforts to target cancers with aberrant expression FGFRs. Clearly, the authors set out to provide a truly comprehensive review. With one exception, they have achieved this goal. However, the authors may want to add a section (even if only a short section) discussing the existence of FGFR splice variants, especially for FGFR1-3, and their roles in different cancers. This is the one area not covered by this excellent, and comprehensive review.

Author Response

Comment1: However, the authors may want to add a section (even if only a short section) discussing the existence of FGFR splice variants, especially for FGFR1-3, and their roles in different cancers.

Response1: We sincerely thank the reviewer for the very positive and encouraging evaluation of our manuscript, as well as for highlighting an important area that was not explicitly addressed in the original version. We fully agree that alternative splicing represents a critical but often underappreciated dimension of FGFR biology.

In response to this constructive suggestion, we have added a new dedicated subsection (Section 1.2.5, Lines 297–317) entitled “Aberrant Alternative Splicing of FGFR1–3 in Solid Tumors.” In this section, we summarize the major cancer‑relevant splice variants of FGFR1–3, with particular emphasis on the well‑characterized IIIb/IIIc isoforms arising from alternative splicing of the IgIII domain, their differential ligand specificity, and their roles in tumor–stromal interactions and signaling plasticity.

We further discuss tumor‑specific examples, including truncated FGFR1β variants associated with enhanced mitogenic signaling, FGFR2 IIIb‑to‑IIIc switching linked to epithelial–mesenchymal transition and invasiveness, and FGFR3 splice variants that alter C‑terminal kinase regulation and therapeutic sensitivity. Collectively, this addition highlights aberrant FGFR splicing as an additional layer of dysregulation in solid tumors, with implications for tumor progression, biomarker interpretation, and therapeutic targeting.

We believe that this new subsection directly addresses the reviewer’s concern and further strengthens the completeness and translational relevance of the review. We are grateful for this insightful recommendation, which has improved the overall scope of the manuscript.

Round 2

Reviewer 2 Report

Comments and Suggestions for Authors

Accept